# Polar bear energetic and behavioral strategies on land with implications for surviving the ice-free period

Anthony M. Pagano [1] ✉, Karyn D. Rode [1], Nicholas J. Lunn [2], David McGeachy [2], Stephen N. Atkinson [3], Sean D. Farley [4], Joy A. Erlenbach[5,7] & Charles T. Robbins[5,6]

Declining Arctic sea ice is increasing polar bear land use. Polar bears on land are thought to minimize activity to conserve energy. Here, we measure the daily energy expenditure (DEE), diet, behavior, movement, and body composition changes of 20 different polar bears on land over 19–23 days from August to September (2019–2022) in Manitoba, Canada. Polar bears on land exhibited a 5.2-fold range in DEE and 19-fold range in activity, from hibernation-like DEEs to levels approaching active bears on the sea ice, including three individuals that made energetically demanding swims totaling 54–175 km. Bears consumed berries, vegetation, birds, bones, antlers, seal, and beluga. Beyond compensating for elevated DEE, there was little benefit from terrestrial foraging toward prolonging the predicted time to starvation, as 19 of 20 bears lost mass (0.4–1.7 kg•day$^{-1}$). Although polar bears on land exhibit remarkable behavioral plasticity, our findings reinforce the risk of starvation, particularly in subadults, with forecasted increases in the onshore period.

Energy is the primary currency that determines individual survival and reproductive success. Animals inhabiting environments with limited food resources may minimize their movements and energy expenditure and rely on stored reserves ('fasting response')[1,2], or increase their movements and energy expenditure as they search for food ('foraging or hunger response')[3,4]. Such responses likely depend upon the amount of energy reserves an individual has, the duration of the reduced resource period, the energetic tradeoffs between the costs of locating food and the potential energy gained, and even individual variation in behavior.

The importance of individual variation in species survival and natural selection has been recognized since Darwin[5], yet inter-individual variation in behavior is an often-overlooked driver of energy balance and population dynamics[6]. One reason such variation is often neglected is the challenge of collecting detailed individual-specific data. Nevertheless, individual variation in behavior manifests itself in multiple forms, including phenotypic plasticity[7], personality[8], and foraging specialization[9], which may be influenced by an individual's age and experience level[10]. Omnivores and other generalist predators often exhibit greater individual variation relative to more specialized predators[11]. This individual variation can influence overall population responses to changes in resource availability with implications for ecosystem dynamics[9].

The Arctic marine ecosystem is experiencing rapid declines in sea ice extent, age, and thickness[12], which are altering ecological dynamics[13]. Within this ecosystem, polar bears (*Ursus maritimus*) are an apex predator that use sea ice as a platform to hunt primarily ringed (*Pusa hispida*) and bearded seals (*Erignathus barbatus*). Polar bears acquire the majority of their energy resources during a brief period in the late spring and early summer when seals are giving birth to and

[1]U. S. Geological Survey, Alaska Science Center, Anchorage, AK 99508, USA. [2]Wildlife Research Division, Science and Technology Branch, Environment and Climate Change Canada, Edmonton, AB T6G 2E9, Canada. [3]226104 Melrose Road, Cooks Creek, MB R5M 0B9, Canada. [4]Alaska Department of Fish and Game, Anchorage, AK 99518, USA. [5]School of Biological Sciences, Washington State University, Pullman, WA 99164, USA. [6]School of the Environment, Washington State University, Pullman, WA 99164, USA. [7]Present address: U.S. Fish and Wildlife Service, Kodiak National Wildlife Refuge, Kodiak, AK 99615, USA. ✉ e-mail: apagano@usgs.gov

weaning their pups[14]. Climate warming is increasing the duration that some areas of the Arctic are ice free, which in turn forces polar bears in these regions to move to land. In western Hudson Bay, the ice-free period has increased by 3 weeks from 1979–2015, keeping bears on land for approximately 130 days during the past decade[15]. While onshore, these bears are thought to primarily fast[16,17] or consume vegetation with limited energetic benefit[18,19], although some individuals have been documented feeding on terrestrial animals[20]. Further increases in the time polar bears are forced onto land where they are unable to hunt blubber-rich, energy-dense seals is likely to negatively impact their body condition, survival, and reproductive success[21].

Model estimates predict a decline in western Hudson Bay polar bear litter size of 22–67% by mid-century if female polar bears are forced on land a month earlier relative to the 1990s[22]. Others have predicted that up to 24% of the adult males would die of starvation if the summer fast increases to 180 days[23–25]. Despite these predictions, it is unclear if polar bears can extend the duration that they can survive on land in the absence of marine-mammal prey by reducing their energy expenditure while on land, whether through behavioral or physiological adaptations, or if use of terrestrial food resources could meet those energy demands[19,26,27].

To resolve these questions, we measured polar bear daily energy expenditure (DEE), diet, behavior, activity, movement, and body composition over 3-week periods in western Hudson Bay. DEE and body composition were measured using doubly-labeled water and isotopic dilution. Global positioning system (GPS)–equipped video camera collars with tri-axial accelerometers were used to determine diet, activity, behavior, and movement rates, which in turn were used

to assess the causes of variation in DEE (Fig. 1). We evaluated factors influencing individual energetic balance using video-derived observations of foraging and measures of blood biochemistry. We predicted that individuals with greater percentages of body fat would exhibit a fasting response and have reduced activity and energy expenditure. Adult males can be up to twice as large as adult females[28]. Hence, we predicted that adult males would have the lowest activity and mass-specific energy expenditure (e.g., fasting response) due to the allometric relationship between body size and fasting endurance[29,30]. We similarly predicted that pregnant females would constrain their activity and energy expenditure (fasting response), albeit to a lesser extent than adult males given their need to seek inland den locations. Pregnant females are known to be in a negative energy balance for up to 8 months between the time they come on land in summer and return to the sea ice the following spring[31]. Conversely, subadult bears may be more likely to exhibit a foraging response if they have not accrued adequate fat stores prior to the start of the onshore period. Given their smaller body size, we predicted subadult females would be more likely to exhibit a foraging response on land compared to subadult males. Lastly, we estimated the time each bear in the study might survive the summer period on land without additional feeding, based upon its initial and final body composition and DEE.

## Results
### Energy expenditure
We measured the energy expenditure, diet, behavior, activity, movement rate, blood chemistry, and body composition of 20 polar bears: 8 solitary adult females, 5 adult males, 4 subadult females, and

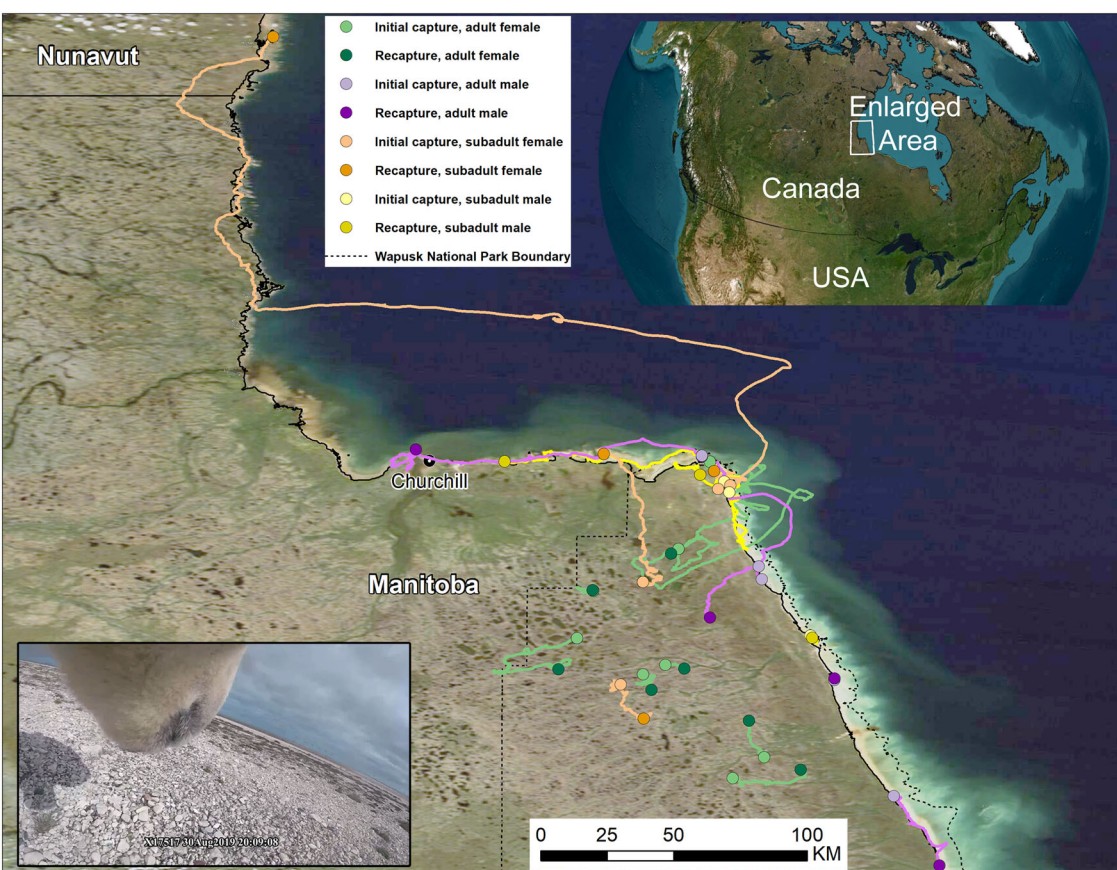

**Fig. 1 | Map of polar bear movements derived from GPS-enabled video camera collars.** Capture (light points) and recapture (dark points) locations and GPS movement paths of 20 polar bears (8 adult females (green lines), 4 subadult females (orange lines), 3 subadult males (yellow lines), and 5 adult males (purple lines)) dosed with doubly-labeled water and equipped with GPS-enabled video camera collars on land near Churchill, Manitoba, Canada. (Inset) Image from a GPS-equipped video camera collar on an adult male polar bear near Churchill, Manitoba, Canada (datetime in GMT). Source data are provided as a Source Data file.

**Table 1 | Individual measures of polar bear energy expenditure, blood chemistry, and changes in body mass and composition**

| Year | Bear | Age (yrs) | Sex | Duration (days) | Initial Mass (kg) | Final Mass (kg) | LBM change (kg) | Fat change (kg) | % Mass Change | Initial RER | Final RER | Initial serum $P_4$ (ng·ml⁻¹) | Final serum $P_4$ (ng·ml⁻¹) | Initial U/C ratio | Final U/C ratio | DEE (kJ·kg⁻¹·day⁻¹) Speakman | DEE (kJ·kg⁻¹·day⁻¹) Nagy |
|---|---|---|---|---|---|---|---|---|---|---|---|---|---|---|---|---|---|
| 2019 | X17517 | 11 | M | 19 | 410 | 387 | -12.3 | -11.2 | -6% | 0.85 | 0.89 | | | 10.4 | 9.1 | 51.6 | 58.7 |
| 2019 | X19911 | 19 | F | 19 | 273 | 242 | -28.8 | -2.2 | -11% | 0.77 | 0.70 | 0.06 | 0.33 | 8.6 | 7.5 | 157.7 | 176.3 |
| 2021 | X33823 | 4 | M | 20 | 235 | 267 | -1.4 | 33.4 | 14% | 0.63 | 0.77 | | | 6.6 | 92.1 | 124.7 | 147.6 |
| 2021 | X33824 | 4 | M | 21 | 215 | 204 | -10.2 | -0.8 | -5% | 0.87 | 0.69 | | | 16.4 | 8.0 | 81.7 | 96.0 |
| 2021 | X33935 | 4 | F | 20 | 182 | 165 | -12.7 | -4.8 | -9% | 0.63 | 0.66 | | | 23.4 | 7.7 | 90.8 | 105.7 |
| 2021 | X33939 | 3 | F | 19 | 202 | 192 | -9.5 | -0.5 | -5% | 0.57 | 0.66 | | | 10.7 | 6.0 | 217.7 | 254.1 |
| 2021 | X33653 | 6 | F | 21 | 165 | 157 | -5.7 | -2.8 | -5% | 0.87 | 1.00 | 0.40 | 0.41 | 7.8 | 5.9 | 130.3 | 173.1 |
| 2021 | X33934 | 10 | F | 21 | 262 | 242 | -6.9 | -12.6 | -8% | 0.79 | 0.77 | 6.09 | 10.46 | 11.3 | 2.8 | 73.7 | 97.3 |
| 2021 | X33928 | 8 | F | 21 | 263 | 244 | -14.3 | -4.2 | -7% | 0.70 | 0.70 | 5.96 | 11.08 | 21.5 | 4.3 | 99.7 | 122.7 |
| 2021 | X33936 | 7 | F | 21 | 209 | 191 | -11.6 | -6.9 | -9% | 0.69 | 0.88 | 4.63 | 7.28 | 10.0 | 7.7 | 105.8 | 134.2 |
| 2021 | X33938 | 7 | F | 21 | 259 | 245 | -14.1 | 0.1 | -5% | 0.78 | 0.68 | 4.69 | 3.59 | 8.3 | 3.7 | 94.6 | 122.9 |
| 2022 | X33991 | 2 | F | 21 | 155 | 139 | 0.7 | -16.7 | -10% | 0.76 | 0.79 | | | 6.9 | 9.6 | 191.5 | 222.1 |
| 2022 | X33954 | 2 | F | 22 | 166 | 150 | 1.5 | -18.0 | -10% | 0.74 | 0.66 | | | 7.7 | 8.9 | 154.9 | 172.2 |
| 2022 | X33851 | 3 | M | 21 | 273 | 243 | -7.2 | -22.8 | -11% | 0.57 | 0.64 | | | 13.1 | 14.3 | 117.7 | 132.1 |
| 2022 | X33410 | 15 | F | 21 | 318 | 305 | 8.2 | -21.7 | -4% | 0.69 | 0.82 | 6.17 | 6.48 | 10.7 | 6.0 | 60.5 | 67.8 |
| 2022 | X33712 | 5 | F | 23 | 296 | 273 | 2.9 | -25.9 | -8% | 0.69 | 0.61 | 5.30 | 11.35 | 3.2 | 3.2 | 104.9 | 117.3 |
| 2022 | X32415 | 21 | M | 21 | 446 | 414 | -18.6 | -13.9 | -7% | 0.92 | 0.84 | | | 6.2 | 7.9 | 41.8 | 47.0 |
| 2022 | X19842 | 21 | M | 23 | 526 | 490 | -14.0 | -22.0 | -7% | 0.73 | 0.65 | | | 4.8 | 2.9 | 72.9 | 82.1 |
| 2022 | X33302 | 17 | M | 21 | 484 | 451 | -4.3 | -28.7 | -7% | 0.72 | 0.73 | | | 4.6 | 9.3 | 78.9 | 87.0 |
| 2022 | X32422 | 18 | M | 21 | 583 | 550 | 11.8 | -45.3 | -6% | 0.55 | 0.65 | | | 10.7 | 18.4 | 164.6 | 186.0 |

Daily energy expenditure (DEE), respiratory exchange ratio (RER), serum progesterone ($P_4$), and changes in total body mass, lean body mass (LBM), and fat mass of 20 polar bears on land near Churchill, Manitoba, Canada in 2019, 2021, and 2022. Estimates of DEE are presented using Speakman's[57] two-pool equation ('Speakman') and Nagy's[61] one-pool equation ('Nagy') for $CO_2$ production. Serum $P_4$ levels > 2.5 ng·ml⁻¹ are considered indicative of pregnancy in polar bears in autumn[65]. Blood serum urea/creatinine ratios (U/C) ≤ 16 are considered to be an indicator of fasting for >1 week[34]. RERs <0.70 are believed to reflect ketogenesis while fasting[58].

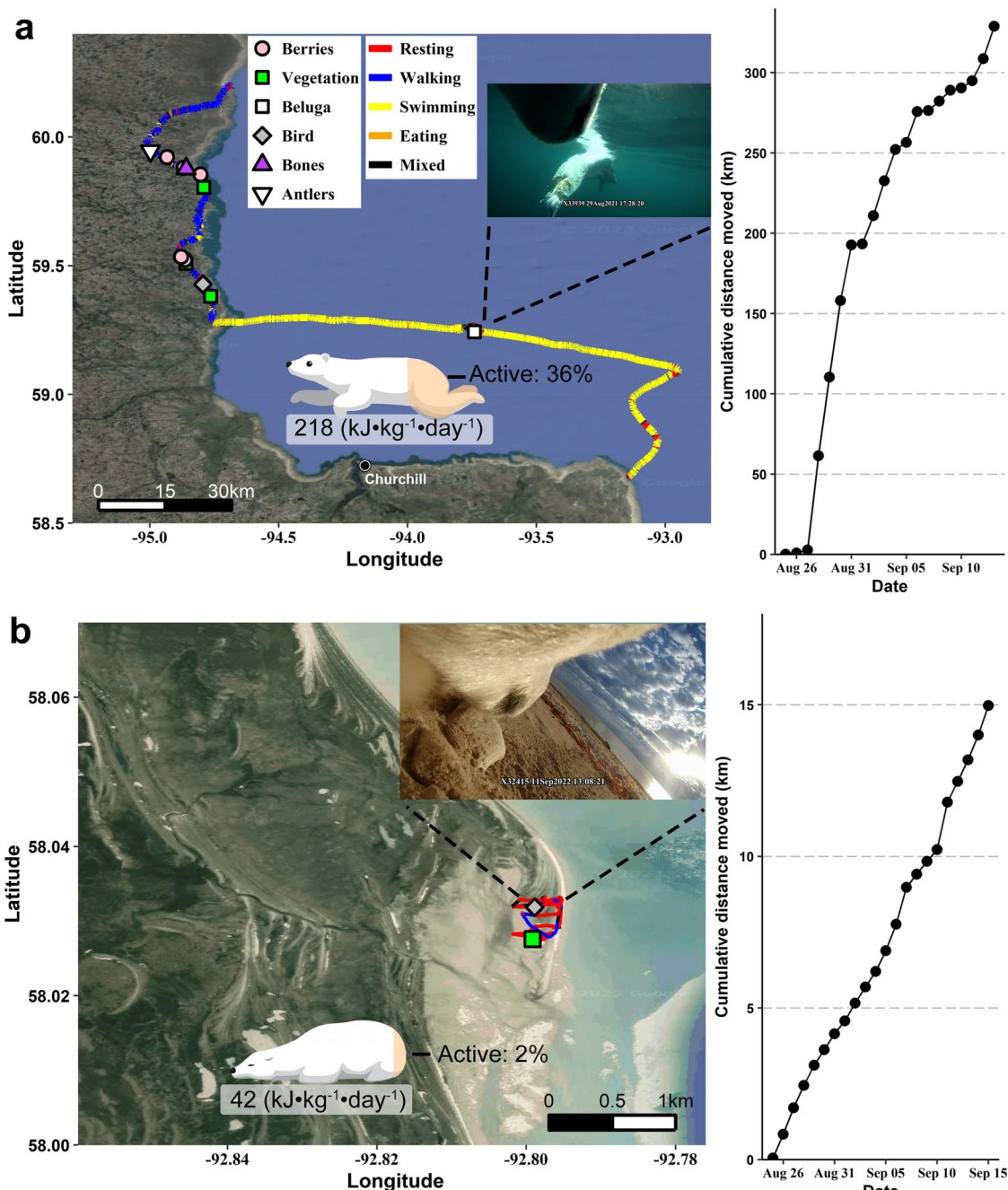

**Fig. 2 | Contrasting movements, activity, behavior, diet, and energy expenditure of a subadult female and an adult male polar bear on land. a** A subadult female (X33939) monitored for 19 days and **b** an adult male (X32415) polar bear monitored for 21 days on land near Churchill, Manitoba, Canada. Movements were derived from GPS location data at 5-min intervals. Movements are color-coded to show where bears spent ≥50% of time resting, walking, swimming, eating, or a mix of these behaviors. Feeding events were derived from the video-camera footage. Behaviors and activity levels were derived from tri-axial accelerometer data. The bear shading (orange) reflects the overall mean activity expressed as a % total time spent active (non-resting). Also shown is the mean daily energy expenditure derived using doubly-labeled water, a representative image derived from the video-camera collar (subadult female: swimming toward a beluga carcass, adult male: resting; datetimes in GMT), and the cumulative daily distance moved derived from the GPS location data. Source data are provided as a Source Data file.

3 subadult males (Fig. 1, Table 1)[32]. Body mass ranged from 147–566 kg ($\bar{x} = 287 \pm 27$ kg, $n = 20$) and total energy expenditure over 19 to 23 days ranged from 313–1956 MJ ($\bar{x} = 618 \pm 79$ MJ, $n = 20$) or 75,000–467,000 kcal ($\bar{x} = 148,000 \pm 19,000$ kcal). Mass-specific DEE varied 5.2-fold among individuals with a range of 41.8–217.7 kJ·kg$^{-1}$·day$^{-1}$ ($\bar{x} = 110.8 \pm 10.5$ kJ·kg$^{-1}$·day$^{-1}$, $n = 20$) (Fig. 2, Table 1). Subadult females exhibited the greatest DEE ($\bar{x} = 163.7 \pm 27.5$ kJ·kg$^{-1}$·day$^{-1}$, $n = 4$), followed by subadult males ($\bar{x} = 108.0 \pm 13.3$ kJ·kg$^{-1}$·day$^{-1}$, $n = 3$), while adult males exhibited the lowest ($\bar{x} = 82.0 \pm 21.8$ kJ·kg$^{-1}$·day$^{-1}$, $n = 5$) (Fig. 3a). Pregnant adult females had lower DEE ($\bar{x} = 89.9 \pm 7.6$ kJ·kg$^{-1}$·day$^{-1}$, $n = 6$) than non-pregnant adult females without cubs ($\bar{x} = 144.0 \pm 13.7$ kJ·kg$^{-1}$·day$^{-1}$, $n = 2$) (Fig. 3a). The coefficient of variation for DEE of all bears was 42%. The coefficient of variation was 29.5 for adult females, 33.6 for subadult females, 21.3 for subadult males, and 59.4% for adult males. Using a multiple linear regression, the relationship between mass-specific DEE was best supported by an additive effect between sex and age class and activity, which explained 72% of the variation in DEE

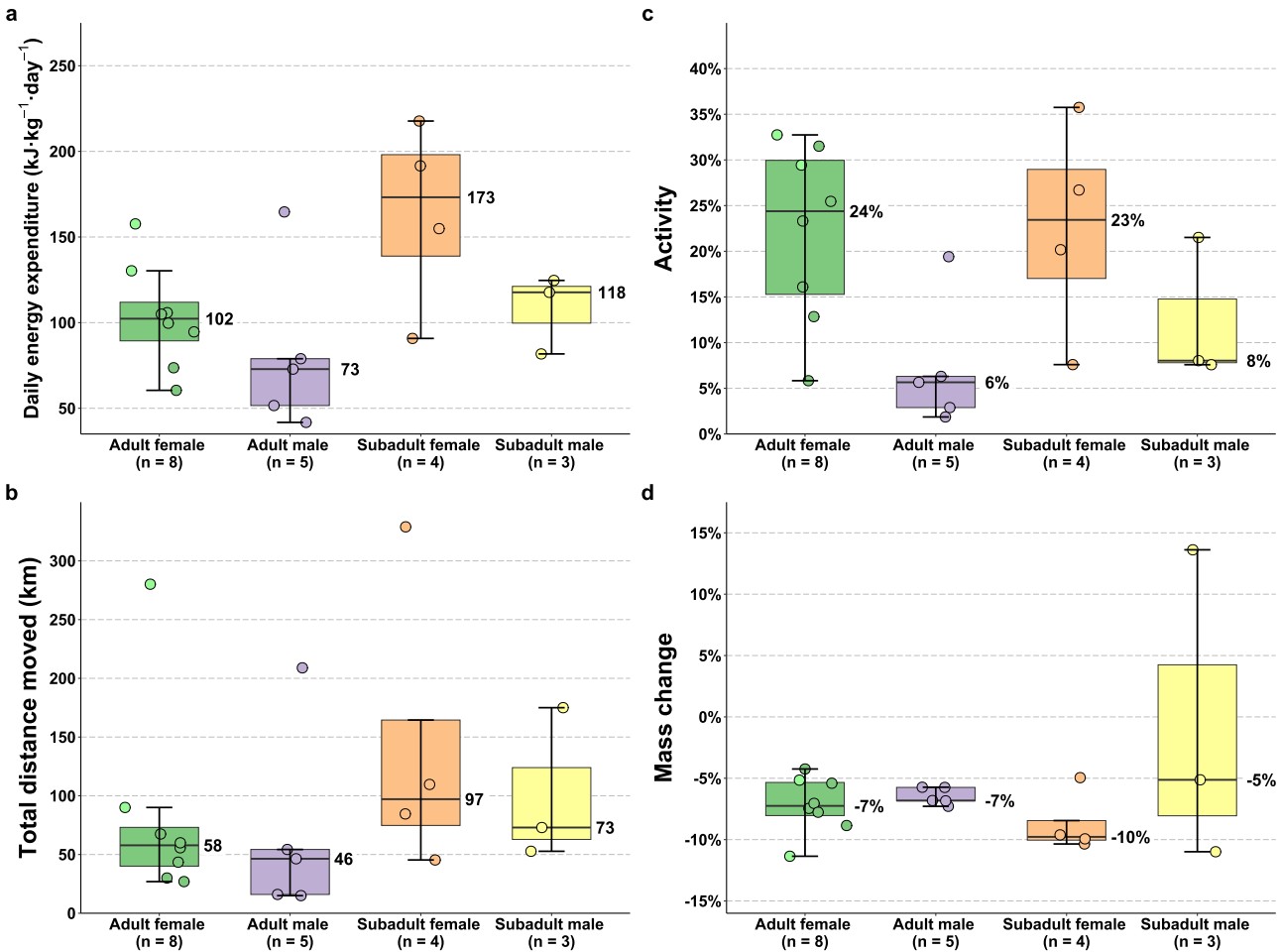

**Fig. 3 | Differences in energy expenditure, movements, activity, and changes in body mass of polar bears on land by age and sex class.** Data are summarized from 8 adult female (6 pregnant (green points), 2 non-pregnant (light green points)), 5 adult male, 4 subadult female, and 3 subadult male polar bears on land near Churchill, Manitoba, Canada. **a** Daily energy expenditure derived using doubly-labeled water, **b** total distance moved based on GPS location data, **c** accelerometer-derived activity expressed as a % total time spent active (non-resting), and **d** percent body mass change between initial capture and recapture 19–23 days later. Boxplots indicate the median, 1st and 3rd quartiles, maximum within 1.5× the inter-quartile range, and minimum within 1.5× the inter-quartile range. Numbers next to boxplots indicate the median values by sex and age class. Points represent raw values per individual. Source data are provided as a Source Data file.

($P < 0.001$) (Fig. 4a, Supplementary Table 1). Variation in activity derived from the accelerometer data better predicted variation in DEE than the percent time swimming or movement rate (Supplementary Table 1). Sex and age class as a categorical variable better predicted variation in DEE than including age as a continuous variable (Supplementary Table 1). Percent initial body fat and the percent body mass change were not supported in predicting DEE (all models ΔAIC$_c$ > 2, Supplementary Table 1).

We found a significant allometric relationship between DEE (MJ•day$^{-1}$) and mean body mass (i.e., the mean body mass between the initial capture and recapture; kg) (DEE = 2.077 × mass$^{0.459}$, $R^2 = 0.27$, DF = 18, $p = 0.05$, Fig. 4b). Two adult males had DEEs less than or equal to predicted mean hibernating rates of Holarctic bears[24], 1 subadult male and 1 subadult female had DEEs within 5 and 10% of Kleiber's predicted basal metabolic rates[33], respectively, and 2 pregnant adult females had DEEs less than or equal to predicted basal metabolic rates (Fig. 4b). In contrast, 70% ($n = 14$) of bears had DEEs 2–4× predicted mean hibernating rates and 40% ($n = 8$) had DEEs 2–4× predicted basal metabolic rates, including 1 subadult female with a DEE within 19% of the predicted mean DEE of female polar bears on the spring sea ice (Figs. 2a and 4b).

## Behavior, diet, and changes in body mass
Total distance moved derived from the GPS location data varied by 22-fold among individuals, ranging from 15–329 km ($\bar{x} = 93 \pm 20$ km) (Supplementary Movie 1). The overall time spent swimming ranged from 0–16% including an adult female that swam a total of 120 km, a subadult female that swam a total of 175 km (Fig. 2), and an adult male that swam a total of 54 km. Subadult females and males exhibited 1.2–2.1× greater movements on average relative to adult females and males (Fig. 3b). Overall activity derived from accelerometer data varied by more than 19-fold among individuals, ranging from 2–36% ($\bar{x} = 17 \pm 2$%) (Figs. 2 and 5). Adult and subadult females exhibited 1.8–3.1× greater activity on average relative to adult and subadult males (Fig. 3c), primarily due to spending more time eating (Figs. 5 and 6, Supplementary Fig. 1).

Despite large variation in activity and DEE, percent changes in body mass were similar across sex and age classes, and 19 of the 20 bears lost 4–11% ($\bar{x} = 7.4 \pm 0.5$%) of their body mass (Fig. 3d, Supplementary Fig. 2, Table 1). One subadult male gained 32 kg (13.6%) between his initial capture and recapture (Fig. 3d, Supplementary Fig. 2, Table 1). Daily energy expenditure received some support in explaining the percent change in body mass (excluding the 1 individual

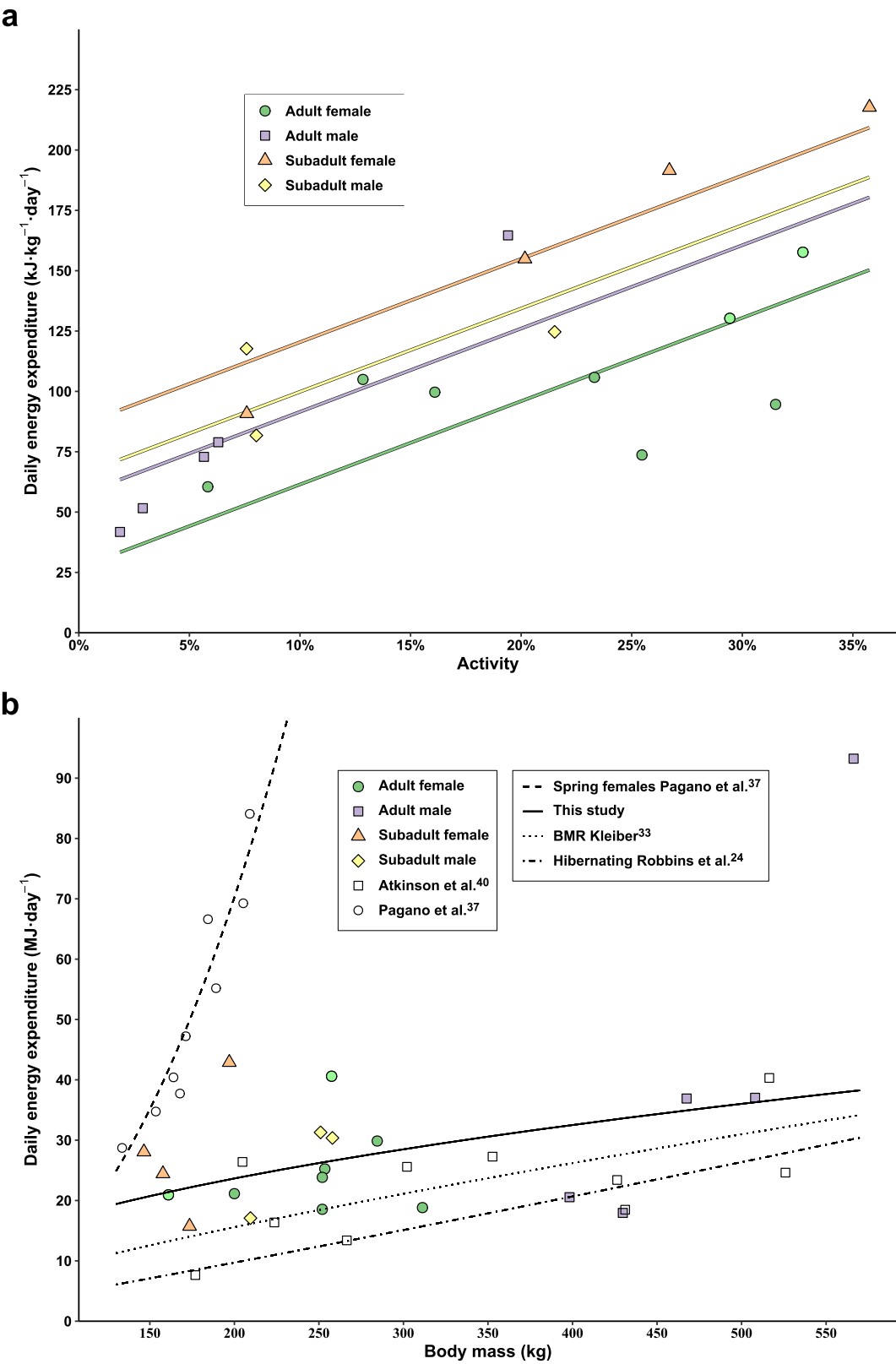

**Fig. 4 | The effects of activity and body mass on the energy expenditure of polar bears on land.** Daily energy expenditure (DEE) of 8 adult female (6 pregnant (green points), 2 non-pregnant (light green points)), 5 adult male, 4 subadult female, and 3 subadult male polar bears on land near Churchill, Manitoba, Canada over 19–23 days. **a** Multiple linear regression in relation to age and sex class and activity derived from tri-axial accelerometer data expressed as a % total time spent active (non-resting). **b** The allometric regression (solid line) of DEE with mean body mass compared to the DEE of female polar bears on the spring sea ice in the Beaufort Sea (white points and dashed line)[37], predicted DEE of male polar bears on land in western Hudson Bay based on changes in body composition (white squares)[40], predicted basal metabolic rates (BMR; dotted line)[33], and the average energetic cost of hibernation in Holarctic bears (dash-dot line)[24]. Source data are provided as a Source Data file.

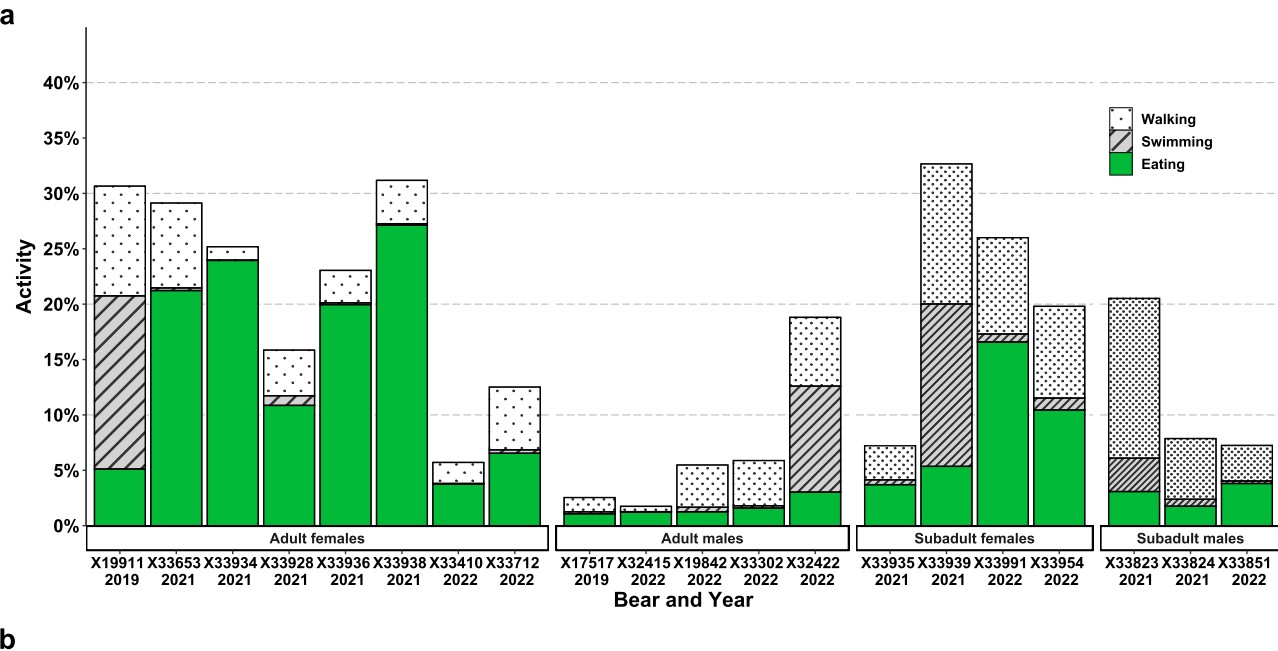

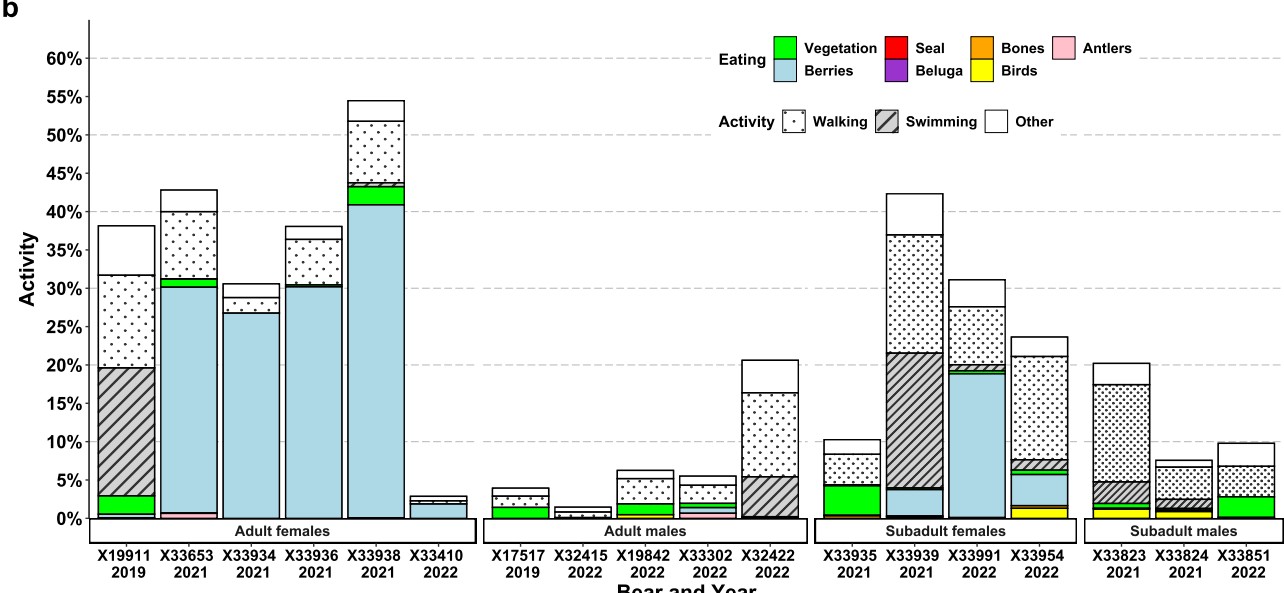

**Fig. 5 | Percent time engaged in active (non-resting) behaviors by polar bears on land near Churchill, Manitoba, Canada. a** Behaviors derived from tri-axial accelerometer data recording continuously at 16 Hz from 20 polar bears, and **b** behaviors derived from video footage from 18 polar bears recorded during daylight hours. Source data are provided as a Source Data file.

that gained mass) (ΔAIC$_c$ = 1.4, Supplementary Table 2), but this relationship only explained 2% of the variation in the percent change in body mass (DF = 17, $p$ = 0.26). None of the other variables explained the variation better than a null model (Supplementary Table 2). Daily change in body mass ranged from −1.7−−0.4 kg•day$^{-1}$ ($\bar{x}$ = −1.0 ± 0.1 kg•day$^{-1}$) for the 19 bears that lost mass, with adult females losing 0.9 kg•day$^{-1}$ (±0.1), subadult females losing 0.7 kg•day$^{-1}$ (±0.1), two subadult males losing 1.0 kg•day$^{-1}$ (±0.5), and adult males losing 1.5 kg•day$^{-1}$ (±0.1).

Video collars recorded a mean total of 6 h (± 20 min, $n$ = 18) of footage during daylight hours, excluding 2 collars that failed within 1- and 4-days post-capture and were excluded from analyses. Overall, 94% of the bears fed on vegetation (grasses and kelp), 56% fed on berries, 39% fed on bird carcasses, 33% chewed on bones, and 17% chewed on caribou antlers (*Rangifer tarandus*) (Supplementary Fig. 1, Supplementary Movies 2, 3). Other prey items were only consumed by single individuals and included bird eggs, a microtine rodent, an Arctic hare (*Lepus arcticus*), seal, and beluga (*Delphinapterus leucas*) (Figs. 2, 5 and 7, Supplementary Fig. 1). The amount of time bears were recorded eating within the duty cycled footage ranged from 1 min to 3 h (Supplementary Fig. 1). Four adult females (3 pregnant) spent a total of 2–3 h of the recorded footage primarily feeding on berries. Despite this high rate of omnivorous food consumption, only two bears at recapture had blood serum urea/creatinine (U/C) ratios indicative of having recently fed (i.e., U/C > 16)[34] (Table 1). Movement rates, activity, and time spent eating were all greater during daylight hours with time spent eating occurring 2–11× more frequently during daylight hours among age and sex classes based on tri-axial accelerometer data (Fig. 6).

The collar of the subadult male that gained body mass stopped recording video 6 days prior to his recapture. Prior to the video failure, he spent 1% of the time feeding on vegetation, 1% feeding on 4 birds, 1% grooming, 1% digging, 3% swimming, 13% walking, and 80% resting (see[32] for behavior definitions) (Fig. 5b, Supplementary Fig. 1,

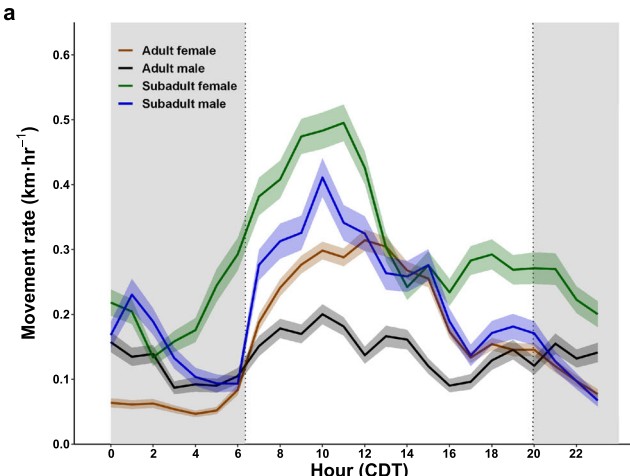

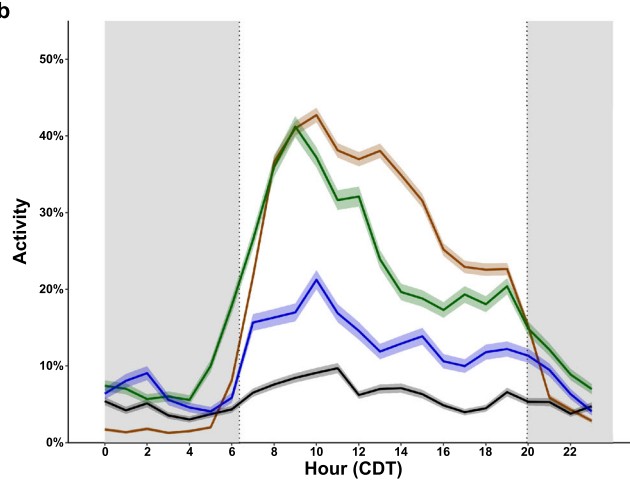

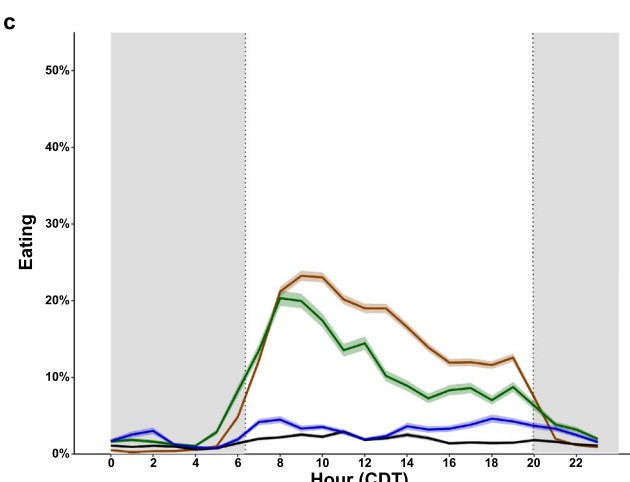

**Fig. 6 | Hourly changes in polar bear activity rates while on land near Churchill, Manitoba, Canada. a** Mean (±SE) hourly movement, **b** mean activity (±SE), and **c** mean eating (±SE) of 8 adult female, 5 adult male, 4 subadult female, and 3 subadult male polar bears. Movement rates were derived from GPS location data. The percent time active (i.e., non-resting) or eating were derived from tri-axial accelerometer data recorded continuously at 16 Hz. Dotted vertical lines are the mean timing of sunrise and sunset based on each bear's GPS locations and gray shaded areas represent dark periods between sunset and sunrise. Times are Central Daylight Time (CDT). Source data are provided as a Source Data file.

Supplementary Movie 3). He also chewed on bones from both a bird carcass and caribou carcass. His locations after the video failure were exclusively along the coast. When recaptured his belly was full and his blood serum urea/creatinine ratio was 92.2, indicative of recent feeding on a large food resource. The substantive mass increase, coastal movements, and large change in urea/creatinine ratio all suggest he likely fed on a seal, beluga carcass, or other large mammal within a few days of recapture.

### Predicted date of starvation
We found no significant difference in the predicted date of death by starvation between the initial capture and recapture (mean difference = −4.8 days, $t$ = −1.2 days, DF = 19, $p$ = 0.3, Fig. 8) based on each bear's body composition and DEE. Sea ice freeze-up in western Hudson Bay averaged 30 November from 2013–2022. Two subadult females had predicted dates of starvation that preceded this date of sea ice freeze-up based on their DEE and body composition at their initial capture and recapture (Fig. 8). The subadult male that gained body mass while onshore initially had a predicted date of starvation that preceded this date of sea ice freeze-up but his predicted date of starvation increased by 62 days as a result of presumably feeding on a large mammal. All other bears were predicted to survive the remainder of the summer onshore period based on their measured body composition and the assumption that they would maintain their DEE.

### Discussion
Although most ursids are opportunistic omnivores, a trait that favors individual variation in foraging strategies, polar bears are highly specialized, feeding almost exclusively on ice-dependent seals[14]. Contrary to an energy conservation strategy that has been predicted for polar bears on land[21,35] (but see ref. 36), we found considerable individual variation in energetic and movement strategies with DEE varying as much as 5-fold and activity varying as much as 19-fold. Even within adult bears DEE varied as much as 4-fold and activity as much as 18-fold. In combination with equivalent measures from female polar bears on the spring sea ice, we found that DEE in non-denning adult polar bears can vary as much as 10-fold between life on the spring sea ice (maximum: 402.1 kJ•kg$^{-1}$•day$^{-1}$ [37],) and on land in summer (minimum: 41.8 kJ•kg$^{-1}$•day$^{-1}$, this study) (Fig. 4b) commensurate with a 20-fold range in mean activity (maximum: 40% in the spring, minimum: 2% in the summer). Such plasticity in activity and metabolism underscores the capacity of polar bears to conserve energy, approaching metabolic rates found in hibernating bears (Fig. 4b). Two adult males in this study had DEEs less than or equal to predicted mean hibernating rates and two adult females had DEEs less than or equal to predicted basal metabolic rates, which suggests they exhibited a fasting response to reduced food accessibility. However, 70% of the bears in this study had DEEs 2–4× greater than predicted mean hibernating rates, which suggests they exhibited a foraging response to reduced food accessibility. Paradoxically and contrary to our predictions, the two adult males with lower initial body mass and body fat were less active and had lower mass-specific DEEs than the larger adult males (Fig. 4b), which implies that energy conservation was a more prominent strategy for adult males in poorer condition. In contrast, two of the pregnant adult females with some of the highest percentages of body fat at their initial capture (47% and 50%) had the lowest movement rates and mass-specific DEEs. As predicted, subadult females exhibited greater DEEs and activity relative to subadult males, likely in part due to their smaller body size but potentially also as a result of their younger age. Although individual age failed to explain differences in DEE, age and experience are likely important factors influencing individual variation in DEE and may in part explain the elevated DEEs we found in some of

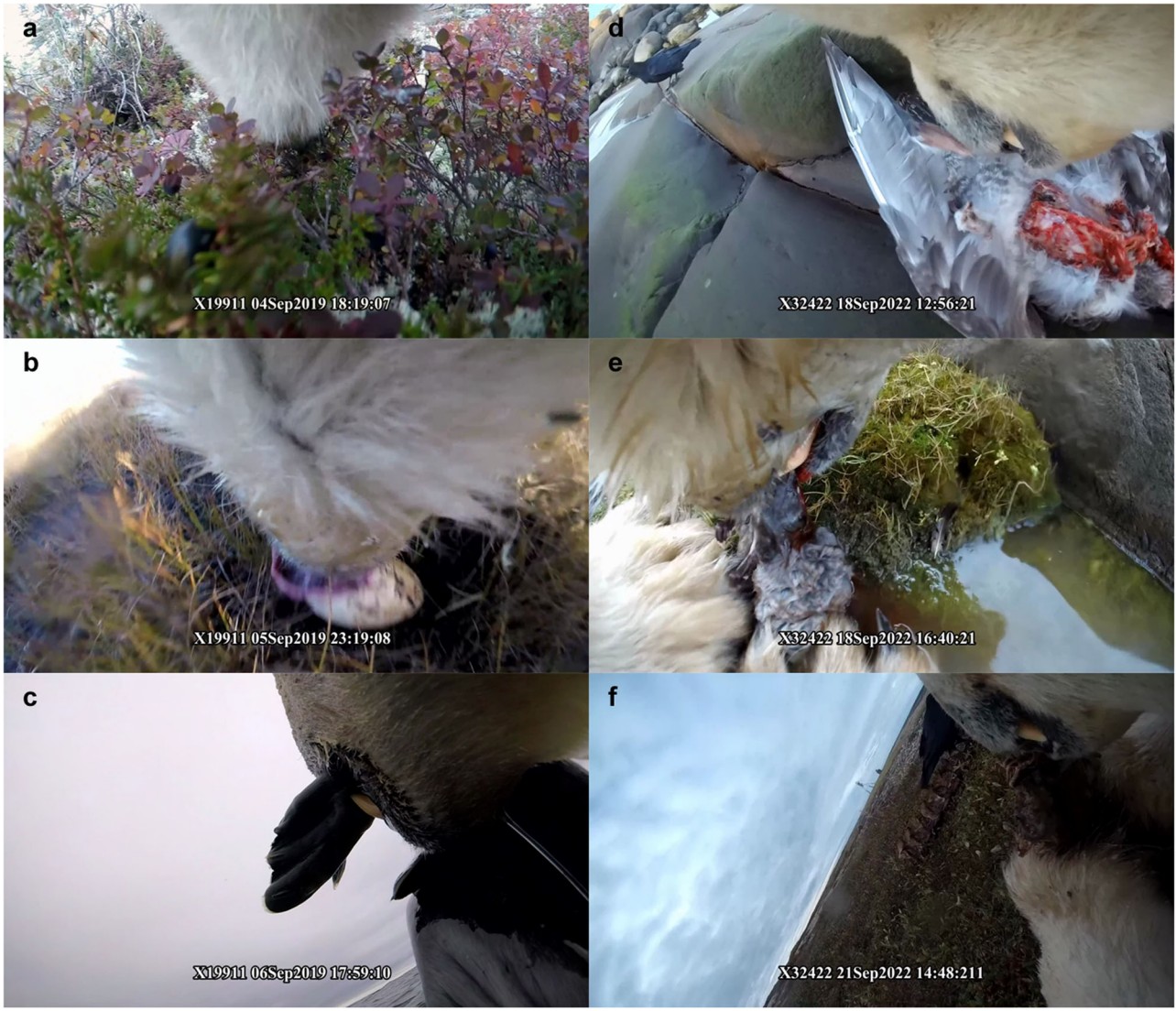

**Fig. 7 | Images from GPS-equipped video camera collars on polar bears on land near Churchill, Manitoba, Canada.** An adult female polar bear eating: **a** berries (*Vaccinium uliginosum*), **b** a waterfowl egg, and **c** holding a seal carcass. An adult male polar bear eating: **d** a gull, **e** a microtine rodent, and **f** chewing on bones from a beluga skeleton. Datetimes in GMT.

the subadult bears that have less experience surviving the ice-free period. Contrary to our expectations, overall, we found no relationship between DEE and percent body fat, which suggests that responses (e.g., activity and DEE) to decreased accessibility of primary prey were largely unrelated to body condition and instead driven by individual-level variation. This variation represents individuals experiencing similar environmental conditions that are responding differently to such conditions regardless of their age, sex class, reproductive status, or body condition[6].

Across age and sex classes we found 42% individual variation in DEEs, which was largely described by differences in activity. Adult and subadult females spent 13% of the time eating on average, primarily consuming berries. Our results are consistent with previous findings of western Hudson Bay polar bears consuming vegetation on land[18,20], but also elucidate the implications of such feeding on activity, DEE, and energy balance. Four adult females (3 pregnant) spent ≥ 20% of daylight hours feeding on berries based on the video-camera collar footage. These berry-feeding bears had DEEs 4% less than four adult females that spent less time feeding on berries, and they experienced only a 0.9% difference in body mass loss (−6.7%) relative to the other four adult females (−7.6%) that spent less time feeding on berries.

Despite having 11% greater activity on average, berry-feeding individuals moved 51% shorter distances than the other four adult females, which likely compensated for their greater activity. Hence, berry feeding by adult female polar bears appears to be an energetically inexpensive strategy albeit with limited energetic benefit.

Three individuals (an adult female, adult male, and subadult female) spent 10–16% of their time swimming. Swimming is energetically demanding on polar bears[38] and long-distance swimming (>50 km) has been assumed to be rare in western Hudson Bay polar bears during the months of our study[39]. However, Pilfold et al.[39] primarily evaluated swimming in adult females with dependent cubs whereas our results suggest it may be a more frequent behavior, particularly in other age, sex, and reproductive classes. Two of these three individuals that swam located carcasses of marine mammals during their swims (Figs. 2 and 7). Yet, the video footage indicates these bears fed minimally from them. The subadult female found a beluga carcass and was only observed feeding on it for 35 s over the 6 h that she was periodically observed near it. Instead, she appeared to use the carcass more as a buoy to rest upon (Fig. 2a). The adult female found a seal carcass and was only observed feeding on it for a total of 20 s over the 7 h she was periodically observed with it. She appeared to attempt to

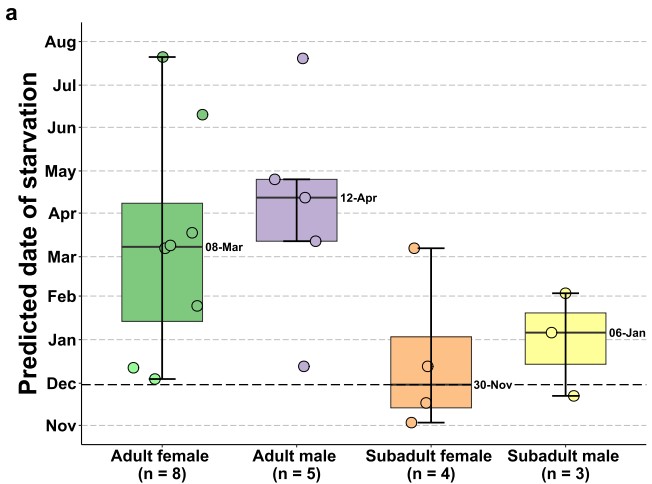

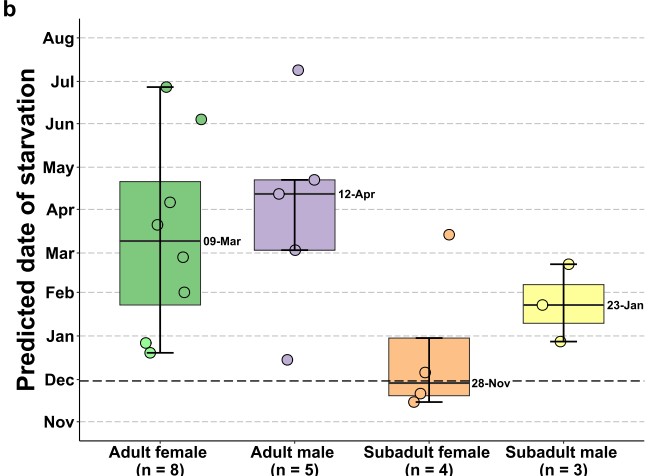

**Fig. 8 | Predicted date of starvation in polar bears on land.** Predicted date of starvation of 8 adult female (6 pregnant (green points), 2 non-pregnant (light green points)), 5 adult male, 4 subadult female, and 3 subadult male polar bears on land near Churchill, Manitoba, Canada based on their body composition and daily energy expenditure relative to the mean date of sea ice freeze up in western Hudson Bay (dark dashed line) 2013–2022. **a** Date of starvation derived from the bear's initial body composition, **b** date of starvation derived from the bear's recapture body composition. Boxplots indicate the median, 1st and 3rd quartiles, maximum within 1.5× the inter-quartile range, and minimum within 1.5× the inter-quartile range. Numbers next to boxplots indicate the median values by sex and age class. Points represent raw values per individual. Source data are provided as a Source Data file.

bring it back to shore before dropping it during her swim. These observations suggest polar bears may be ill-equipped to feed while in water. Consistent with the high energetic costs of swimming in polar bears, these three bears had the highest mass-specific DEEs and the earliest predicted time of starvation of their respective age and sex class; they were also the most active individuals within their respective age and sex classes (Fig. 4a).

Despite large individual variation in DEE, activity, movements, behavior, and diet, changes in percent body mass loss (excluding the subadult male that gained mass) were similar among individuals and sex and age classes. Estimates of daily mass loss were also consistent with previous estimates for western Hudson Bay polar bears while onshore[40]. Pilfold et al.[25] reported median mass loss rates of fasting-detained (i.e., kept in a holding facility a median of 17 days) polar bears of 1.4, 1.0, 1.0, and 0.9 kg•day$^{-1}$ in adult males, solitary adult females, subadult males, and subadult females, respectively, which is similar to the rates of mass loss in our study (1.5, 0.9, 1.0 and 0.7 kg•day$^{-1}$,

respectively). Hence, despite the elevated activity, movement, and food consumption of most bears in our study, mass loss rates were commensurate with rates of mass loss in fasting and relatively inactive bears. This suggests that the more active bears in our study were able to compensate for their elevated DEE through consumption of terrestrial foods. This was further highlighted by our finding that the group of seven bears that lost ≤6% of their body mass, included only two of the five bears with the lowest DEEs but also two of the five bears with the highest DEEs within their respective age and sex classes. Given that most of the bears exhibited a foraging response, it is likely that this apex predator will increasingly influence the terrestrial ecosystem, especially if the timing of arrival on land progressively overlaps with the nesting phenology of waterfowl[41]. Nevertheless, terrestrial foods did not stem the overall rate of mass loss of polar bears on land or significantly alter their predicted time to starvation, further reinforcing the conclusion that terrestrial resources in this region and season are inadequate to prolong the period that polar bears can survive on land[19]. In contrast, the one individual that likely fed on a large mammal on land, increased his body mass by 14% (Supplementary Fig. 2), which is similar to the percent mass gain found in female polar bears feeding on ringed seals on the sea ice over 8–11 days[37]. This highlights the disparity in the energetic windfall polar bears acquire through energy-dense marine mammals relative to terrestrial-based resources[19,37].

Bears in a long-term fast such as hibernation primarily metabolize body fat and experience minimal changes in lean body mass[42,43]. Yet, half of the bears in this study lost more lean body mass than body fat (Table 1, Supplementary Table 3, Supplementary Fig. 2). These findings are consistent with previous research that evaluated changes in the body composition of 10 male polar bears on land in western Hudson Bay and found 60% of those bears lost greater amounts of lean body mass than body fat[40]. The apparent paradox of enhanced lean mass loss compared to fat loss by some bears could be due to the metabolic fuels available for energy provision. Ketones produced by the liver from fat metabolism increase during fasting, and serve as an energy supply for skeletal muscle, heart, and brain. Ketones are also anti-lipolytic and suppress appetite[44]. When hibernating bears are fed glucose, circulating ketones are suppressed to levels seen in the active season[45]. Thus, polar bears feeding on terrestrial foods would likely have suppressed ketone production, resulting in increased appetite and food-seeking behavior. However, the increased energetic cost of foraging when coupled with the increased ketone and glucose uptake by active muscles[44] would likely reduce lipolysis and exacerbate lean mass loss over fat loss. Of the bears that lost more lean body mass than body fat in our study, 50% (i.e., 5 bears) had activity rates > 25%, which suggests that they exhibited a foraging response to reduced food accessibility. Yet, the remainder had lower activity rates including 2 adult males that had activity rates <5%. An alternative mechanism of this increased use of lean body mass over fat mass may be to preserve fat to aid thermoregulation in the fall and winter[46]. Lean body mass is also more energetically costly to maintain than fat mass, which may also favor the increased metabolism of lean body mass in non-hibernating bears in a negative energy balance[47]. Hence, changes in body composition of active polar bears while on land appear to be complex and warrant further research given the implications for overall body condition, energy storage, and predicted time to starvation[21].

Although we were able to measure the ecophysiology of four age and sex classes of polar bears while on land, this resulted in small sample sizes for each age and sex class which in combination with the large individual variation we documented, limits our ability to generalize predictions of polar bear energy demands on land among age and sex classes. Additionally, we did not collect data from adult females with dependent young. Lactation is the most energetically expensive component of the reproductive cycle in mammals[48,49] and can more than double energy demands[48,49]. Hence, we would expect

adult female polar bears with dependent young to have significantly greater DEEs relative to the solitary adult females in our study depending on the extent to which they continue to lactate[50]. While polar bears may nurse their cubs for up to 2 years[51], they appear to be able to reduce lactation investment in conjunction with increased time on land[52]. How they behaviorally respond to such energetic demands warrants further research, particularly to better understand the implications of increased land use on cub survival.

Research on survival, reproductive success, and other fitness metrics typically examine population-level responses with limited focus on individual-level variation in behavior. Our results highlight the potential individual variation that can occur in large apex predators responding to reduced access to primary prey, which may complicate model predictions of populational responses to habitat change without consideration of such variability (e.g., ref. 21). Specifically, our results indicate that while on land, polar bears are not responding uniformly in their behavior, diet, or energetic responses even within age, sex, reproductive classes, or among similar levels of body condition. Nevertheless, declines in body mass were consistent among 95% of the bears in our study, which emphasizes that none of the energetic strategies were more beneficial for surviving the on-land period, though the foraging response may result in opportunistic feeding events on large mammals. Such opportunistic events have been hypothesized to have enabled polar bears to survive past interglacial periods but are predicted to be less of a resource during the current Anthropocene due to lower abundances of whale populations[53]. Ultimately, our findings reinforce the risk of starvation for polar bears on land with forecasted increases in the onshore period.

## Methods

Data were collected from polar bears captured on land in Wapusk National Park, Manitoba, Canada (Fig. 1, Supplementary Movie 1). Polar bears were located from a helicopter and captured using standard chemical immobilization techniques[54]. Bears were sampled between 26 August – 14 September 2019, 25 August – 18 September 2021, and 24 August – 21 September 2022. Bears that had not been previously captured were aged based on counts of cementum annuli from an extracted vestigial premolar. We classified adults as ≥5 years old and subadults as independent bears that were 2–4 years old. All procedures were approved by the Animal Care and Use Committees of the U.S. Geological Survey, Alaska Science Center and the Environment and Climate Change Canada Prairie and Northern Region. Research was approved under permits from Parks Canada Wapusk National Park Research and Collection Permits (#WAP-2020-37418, WAP-2020-36578), Manitoba Species at Risk/Wildlife Scientific Permits (#SAR21014, SAR20021), and an Exemption Permit issued by the Department of Environment, Nunavut. Sample import into the United States was authorized under Marine Mammal Research Permits (MA82088B-1, MA690038-17).

### Doubly-labeled water

Following immobilization, we inserted either an external jugular or cephalic catheter to facilitate blood sampling and administration of isotopes. We took a blood sample at the time of the initial capture to serve as a baseline measure of oxygen-18 ($^{18}O$) and deuterated water ($^{2}H$). The bear was then injected intravenously with a precisely weighed dose containing 0.26–0.64 g·kg$^{-1}$ of 98.4% enriched $^{18}O$ (Isoflex USA, San Francisco, CA) and 0.13–0.32 g·kg$^{-1}$ of 99.8% enriched $^{2}H$ (Sigma Aldrich, Inc., St. Louis, MO) with NaCl added to make it 0.9% isotonic and sterilized using a 0.2 µ Millipore filter (Corning, Inc., Corning, NY) (Supplementary Table 4). On injection, the syringe was backwashed with blood three times to ensure all the doubly-labeled water had been injected into the bear. The bear was kept immobilized for 2 h after the injection of doubly-labeled water to allow isotope equilibration[55,56]. We collected serial blood samples 30, 60, 90, and 120 min after dosing to

evaluate equilibration curves[57]. The bears were weighed using an electronic load cell suspended from an aluminum tripod. We recaptured bears 19–23 days after initial capture to obtain a blood sample to measure final enrichment. Similar to the initial capture, following immobilization, we inserted either an external jugular or cephalic catheter to facilitate blood sampling and administration of isotopes. An initial blood sample was collected to measure final enrichment after which bears were dosed with 0.10–0.11 g·kg$^{-1}$ of 99.8% enriched $^{2}H$ (Sigma Aldrich, Inc.) made isotonic with 0.9% NaCl and sterilized using a 0.2 µ Millipore filter. On injection, the syringe was backwashed with blood three times to ensure all the $^{2}H$ had been injected into the bear and serial blood samples were collected 30, 60, 90, and 120 min after dosing. Blood was collected in 10 ml glass evacuated tubes without anticoagulants (Serum Vacutainer, Becton Dickinson, Franklin Lakes, NJ or Monoject noncoated tubes, Cardinal Health, Dublin, OH) and centrifuged to separate serum from red blood cells. Serum was stored frozen in 2 ml cryogenic vials (Corning, Inc.) at −80 °C until analysis.

We measured the respiratory exchange ratio of bears from respired samples collected by placing a mask (Smiths Medical Inc., Dublin, OH) over the snout of each bear. The respiratory exchange ratio is the ratio of $CO_2$ produced to $O_2$ consumed and can indicate the oxidation of lipids (0.70), proteins (0.82), carbohydrates (1.00), or a mixture of nutrient types (0.70–1.00)[57]. Respiratory exchange ratios <0.70 have been reported in polar bears on land and appear to reflect ketogenesis while fasting[58]. The mask was attached to a two-way valve with ports for inhalation and exhalation (Hans Rudolph Inc., Shawnee, KS), which allowed the bear to inhale fresh air while exhaling into a 25-liter Douglas bag (Harvard Apparatus, Holliston, MA). We calculated the respiratory exchange ratio from these breath samples using an $O_2$ and $CO_2$ analyzer (FOXBOX; Sable Systems International, Las Vegas, NV). We zeroed the $CO_2$ sensor with nitrogen gas and spanned the $CO_2$ sensor with a known concentration of $CO_2$ gas and the $O_2$ sensor with ambient air using Drierite (W. A. Hammond Drierite, Xenia, OH) to remove moisture. We flowed the breath samples through the sensors at 200 ml·min$^{-1}$, preceded and followed by baselines of ambient air. We calculated respiratory exchange ratios using equations 11.7 and 11.8 from Lighton[59] for flow-through respirometry using excurrent flow. We corrected VCO$_2$ measures due to the gas permeability of Douglas bags by adding 0.006 hr$^{-1}$ between the time of collection and analysis (range: 5–14 h)[58].

Water was extracted from each serum sample by vacuum sublimation. Specific activity of $^{18}O$ and $^{2}H$ were determined by wavelength-scanned cavity ring-down spectroscopy (Metabolic Solutions, Inc., Nashua, NH). We calculated $CO_2$ production using the plateau method and Speakman's[57] two-pool equation, which has been shown to be best suited for large mammals[57,60]. Unless otherwise indicated, the DEE reported in the main text was derived using this equation. However, this equation does not adjust for changes in water space, which may occur in animals losing mass. Hence, we also report DEE using Nagy's one-pool equation for calculating $CO_2$ production[61] (Table 1). For Speakman's two-pool equation, we used the mean group dilution space ratio in calculating $CO_2$ production[57]. For both equations, we converted $CO_2$ production to metabolic rate using the mean of the individual-specific respiratory exchange ratio values from the initial capture and recapture and the equation from Weir[62]. We used results from deuterium dilution to evaluate the body composition of bears at initial capture and recapture[40,56]. The total body water estimate was divided by a factor of 1.04 to correct for the non-exchange of deuterium in the body[63]. Measures of fat, lean body mass, and protein were determined from estimates of total body water based on the equations of Farley and Robbins[56], which can have a standard error of ±1.6 kg. For consistency with previous research[37,40] the body composition measures reported in the main text were derived using these equations. However, separate equations have been developed for converting measures of total body water to measures of fat, lean body

mass, and protein in polar bears[64]. Hence, we also report these measures in the supplementary information using these separate equations (Supplementary Table 3).

## Blood chemistry

Serum progesterone ($P_4$) levels of adult female polar bears were measured by radioimmunoassay (ImmuChem™ Progesterone $^{125}$I, MP Biomedicals, LLC, Santa Ana, CA) by either the Endocrine Service Laboratory at the University of Saskatchewan (Saskatoon, SK) or the Endocrinology Laboratory at the Animal Health Diagnostic Center (Cornell University, Ithaca, NY). Previous research has shown serum $P_4$ levels >2.5 ng•ml$^{-1}$ are indicative of pregnancy in polar bears in autumn[65].

Serum samples of all bears were analyzed for blood urea nitrogen (BUN) and creatinine using an Abaxis Vetscan VS2 chemistry analyzer (Abaxis, Inc., Union City, CA). BUN values were divided by 0.466 to obtain urea concentrations[66] and then divided by creatinine concentrations to obtain urea/creatinine ratios (U/C). Urea/creatinine ratios ≤16 were considered indicative of fasting for >1 week prior to capture or recapture[34].

## Video collars and accelerometers

We deployed Global Positioning System (GPS)-enabled video camera collars (Vertex Plus collar with camera option, Vectronic Aerospace GmbH, Berlin, Germany) on the same individuals that were dosed with doubly-labeled water. Duty cycles and schedules of video cameras varied among years with cameras turning on for 10 s every 5 min in 2019 and 5 s every 2 min in 2021 and 2022 (Supplementary Table 5). Collars recorded a GPS fix every 5 min, which were stored in the collar's nonvolatile memory and downloaded upon recovery. Collars also transmitted a subset of these fixes via the Iridium satellite system. We calculated minimum distance traveled between two successive locations as the great-circle distance (i.e., distance accounting for the earth's curvature), and calculated movement rate by dividing distance by the duration between recorded locations (i.e., 5 min) in SAS (version 9.4, SAS Institute Inc., Cary, NC). Additionally, collars measured tri-axial acceleration at 16 Hz (range ±4 g). We calculated whether bears were resting, walking, swimming, or eating based on the accelerometer data using a random forest machine-learning algorithm in R (v. 4.3.2)[67] as described by Pagano et al.[68]. Polar bears on land in western Hudson Bay reach normal movement patterns within 3 days following immobilization[69]. Hence, if anything we would expect our activity and movement patterns to be somewhat biased low due to the inclusion of this potential recovery time in analyses. To visualize spatial changes in behavior, we linked accelerometer-derived behaviors with corresponding GPS location data by calculating the percent time spent in each behavior class between predicted locations (i.e., 5 min) in SAS[70]. We treated the dominant behavior (e.g., ≥50%) between each location as indicative of the primary behavior at the ensuing location. We plotted behavior-linked locations using the R packages 'ggplot2' and 'ggmap'. We used the R package 'maptools' to link GPS coordinates with the timing of sunrise and sunset to examine diurnal and nocturnal differences in activity, movement rates, and eating.

## Analyses

Individual variation across all bears and within sex and age classes was estimated based on the coefficient of variation. We used multiple linear regression with the mass-specific daily energy expenditure (DEE) as the response variable and the percent time active (i.e., inverse of resting) or swimming derived from the tri-axial accelerometer data, movement rate derived from the GPS location data, percent initial body fat, percent body mass change, whether bears were pregnant based on their serum progesterone, and sex and age class as predictors. We similarly used multiple linear regression with the percent body mass change as the response variable and activity (i.e., inverse of resting), swimming, and eating derived from the tri-axial accelerometer data, mass-specific DEE,

movement rate derived from the GPS location data, percent initial body fat, whether bears were pregnant, and sex and age class as the predictors. We assumed activity and movement rate would be collinear and only included them in separate models. Additionally, variance inflation factor (VIF) tests indicated DEE, activity, swimming, and eating were collinear (VIF > 3). Hence, we only included these variables in separate models. Sex and age class was treated as a categorical variable (i.e., adult female, adult male, subadult female, subadult male), pregnancy was treated as a binary variable, and all others were treated as continuous variables. To evaluate whether a bear's experience level influenced its DEE we included age as a continuous variable and evaluated whether age with sex as a binary variable was better supported than sex and age class treated as a categorical variable. We tested for both additive effects and interaction effects between variables and sex and age class to evaluate whether relationships with DEE or percent body mass change differed between sex and age classes. Within these guidelines, we generated a priori sets of candidate models and used an information-theoretic approach based on Akaike's information criterion corrected for small sample size (AIC$_c$). We considered top-ranked models as those with ΔAIC$_c$ values < 2.0. We additionally performed post-hoc tests to evaluate whether any of our continuous variables better described the variation in DEE or percent body mass change as quadratic terms. However, none of these relationships were supported based on AIC$_c$ rankings.

We also evaluated the allometric relationship between individual DEEs and mean body mass in the bears in this study. We estimated differences between DEE relative to body mass from the bears in this study compared to adult and subadult female polar bear DEEs on the sea ice in the spring[37], predicted DEEs of hibernating Holarctic bears[24] where DEE (kcal•day$^{-1}$) = 7.2 × mass$^{1.09}$, and predicted basal metabolic rates[33].

Lastly, we predicted the time to death by starvation for each bear based on its body composition at initial capture and recapture and its average DEE. Similar to Robbins et al[24]. we assumed all total body fat and 30% of total body protein were available as storage energy sources subsequent to starvation. We converted total body fat and total body protein to their energetic equivalents assuming 39.3 MJ•kg$^{-1}$ total body fat and 18.4 MJ•kg$^{-1}$ total body protein[24,40,71]. We divided the sum of each individual's estimated storage energy at their initial capture and recapture by their DEE to predict the date their storage energy would be exhausted in the absence of any food consumption. Similar to Lunn et al.[72], we calculated the date of sea ice freeze-up from 2013–2022 as the first ordinal date in autumn when sea ice concentration within the western Hudson Bay management zone reached and remained above a mean of 50% for at least three consecutive days. We derived sea ice concentration using 25 × 25 km resolution passive microwave satellite data obtained from the National Snow and Ice Data Center[73]. We compared the predicted date of starvation to the mean date of sea ice freeze-up to predict the potential for starvation if individuals were to maintain their observed DEE. We tested whether estimates of the predicted date of starvation differed between the initial capture and recapture using paired t-tests. A significant increase could indicate terrestrial foraging between captures prolonged the time to starvation whereas minimal changes in predicted date of starvation would indicate that changes in body composition were primarily driven by DEE rather than diet. This analysis assumes bears maintained their measured DEE while on land and does not account for potential subsequent feeding events post-recapture that could extend their date of starvation. Additionally, this analysis does not include whether pregnant females had sufficient body reserves left to then enter a maternity den and successfully produce cubs nor does it predict their subsequent DEE while denning. All analyses were conducted in R with α = 0.05. Means are presented ±1 SE.

## Reporting summary

Further information on research design is available in the Nature Portfolio Reporting Summary linked to this article.

## Data availability

All data generated in this study have been deposited and made publicly available through the U. S. Geological Survey Alaska Science Center data repository[32] at: https://doi.org/10.5066/P9A7ITFH. Source data are provided with this paper.

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

## Acknowledgements

This research was supported by the National Science Foundation Office of Polar Programs grant number 2019369 (C.T.R.), Environment and Climate Change Canada (N.J.L. and D.M.), U.S. Geological Survey (A.M.P. and K.D.R.), Churchill Northern Studies Centre, U.S. Fish and Wildlife Service Marine Mammals Management (K.D.R.), San Diego Zoo Wildlife Alliance (A.M.P.), Detroit Zoological Association (K.D.R.), Polar Bears International (A.M.P.), Bureau of Land Management (K.D.R.), and Washington State University (A.M.P. and C.T.R.). We thank helicopter pilot Justin Seniuk (Prairie Helicopters, Inc.) for field support. We thank Daniel Costa for advice on doubly-labeled water analyses and Heiko Jansen for insight on mechanisms driving changes in body composition. We thank Gregory Thiemann for comments on a previous draft of the manuscript. This research used resources of the Core Science Analytics and Synthesis Applied Research Computing program at the U.S. Geological Survey. Any use of trade, firm, or product names is for descriptive purposes only and does not imply endorsement by the U.S. Government.

## Author contributions

A.M.P., K.D.R., N.J.L., D.M., J.A.E. and C.T.R. conceived the study. A.M.P., K.D.R., N.J.L., D.M., S.N.A. and J.A.E. performed the data collection. A.M.P. analyzed the data and wrote the manuscript with assistance and discussion from all the authors.

## Competing interests

The authors declare no competing interests.
