## [Peer Review File · Nature Communications]

Polar bear energetic and behavioral strategies on land with implications for surviving the ice-free periodReviewers' Comments:

Reviewer #1:

Remarks to the Author:

An extremely well-designed study that significantly advances our understanding of polar bear ecology and their fasting abilities. One of the most profoundly interesting studies I've read recently. The presentation of the study is very clear. I have some specific comments for consideration but nothing overly significant.

19 – “Declines in Arctic sea ice are increasing polar bear land use.” many people won't understand the use of land by polar bears as a habitat when sea ice is unavailable. Consider clarifying this statement as many people may only read the abstract. Further, “land use” is a bit vague. It may be useful to state that more bears are using land (across their range) but also for longer. Optional change.

35-7 – “and even individual variation” – isn't the amount of energy an individual has stored also individual variation. Some clarification on what you mean on this point would be useful.

39-40 – “inter-individual variation is an often-overlooked driver of animal behavior, energy balance” I'm unconvinced by this statement. Inter-individual variation is a commonly examined metric for many studies (e.g., age, sex, reproductive status, condition, horn size). Some rewording / refinement needed here.

44 – “increased” vs. “higher” word choice.

45 – I'm a bit uncertain about the “Such variation” – It could be variation within omnivores and generalist predators or the more specialized predators or both collectively. I suspect that latter but because the preceding sentence is set up to show “variation” between the 2 stated groups, consider rewording for clarity. I had to ponder this sentence but couldn't settle on the intended meaning.

51 – “relatively” – relative to what?

52 – I'm not convinced that it is weaning of pups that is the trigger for energy reserves – it is the pupping season, weaning, and mating season for seals. It's more than just the weaning of pups.

68-88 – This section and the predictions is very nicely presented. Concise and well considered.

109 – optional but I find the “respectively” unhelpful. I'd rather see 29.5% for adult females, 33.6% for subadult females... It's easier to keep the 2 parts together.

I may have missed it, but I can't see the study period in the methods nor Table 1. Did the date of sampling affect DEE? I can imagine a bear moving more later in the ice-free period due to migration behavior and not necessarily an individual variation aspect. Specifically, you mention 3-week periods (line 69) suggesting not all sampling of the study animals was concurrent. Please clarify. Somewhere, describe the dates of 1st and last handlings.

116 – because many aspects of polar bear life history are non-linear (i.e., quadratic) did you try a squared term? Mass, litter size, reproductive success and other factors are quadratic which may lead to poor fit.

119-20 – “We found a significant allometric relationship between DEE (MJ•day⁻¹) and mean body mass (kg)” I think mean body mass is probably the mean of the mass at capture and recapture? Please clarify.

Table 1. The second column shows a Bear identifier but I cannot see that this is ever used in the reporting of results. Consider removing this column as it allows focus on the relevant elements. Optional change.

Figure 2a & b. The bear is showing up on my copy as partly white & partly in a pale orange. May be a PDF file issue.

211-13 – Speculation of feeding on a seal or beluga is reasonable but it's also possible that the bear fed on a caribou or another bear. There are even a few walrus in southern Hudson Bay (or dead bowheads). Cannibalism is an option. I would not be so conclusive – perhaps adding “or other large mammal” or just stating a large mammal was likely fed on. This point is raised again on line 324 “likely fed on a marine mammal on land” – again, speculative. Not a big deal but it is perhaps unnecessary. Especially within the context of “windfall” and the specifics of “marine mammal prey”. Further, on line 327, do you really mean “prey”. I don't think of scavenged animals necessarily as “prey” within the context of predation. I didn't see any support for predation vs. scavenging carrion. I find the conclusive nature of the statement unsupported. Same issue on line 369. It is the opportunistic feeding that matters – is it really only marine mammals? A dead moose, caribou, or another polar bear all provide significant energy sources.

299 – wording “ill-equipped” – perhaps “poorly adapted”. Optional.

308-11 – for context the mass loss rates reported in this manuscript and cited in Pilfold et al. are very similar to Derocher, A.E., and Stirling, I. 1995. Temporal variation in reproduction and body mass of polar bears in western Hudson Bay. *Can. J. Zool.* 73(9): 1657-1665. doi: 10.1139/z95-197.

The reported mass loss rates in this paper suggest that mass loss rates have not changed much over time. This is not a point made in the paper but possibly worth noting.

352-4 – Consider reviewing Miller et al. 2022 (*Polar Biology*) who compared starvation threshold of lactating and non-lactating females in the same population. This study supports the “we would expect adult female polar bears with dependent young to have significantly greater DEEs relative to the solitary adult females in our study depending on the extent to which they continue to lactate.” And the vulnerability of nursing females. Optional.

360-60 “Research on survival, reproductive success, and other fitness metrics typically examine population-level responses with limited focus on individual-level variation.” It may be just wording or how one view's research but I think that many studies on survival and reproductive success focus on individual traits (variation between individuals). This sort of research often focusses on variation in age, horn size, antlers, body mass, or other individual traits. I don't really agree with the statement. There are hundreds if not thousands of studies on a huge diversity of taxa from guppies to lions.

302 – the estimation of starvation threshold was calculated for polar bears in this population in this paper Miller, E.N., Lunn, N.J., McGeachy, D., and Derocher, A.E. 2022. Autumn migration phenology of polar bears (*Ursus maritimus*) in Hudson Bay, Canada. *Polar Biol.* 45: 1023-1034. doi: 10.1007/s00300-022-03050-3.

This might be a useful reference for comparison.

The methods and analyses are clearly presented and I have few issues.

409 – The respiratory exchange ratio is outlined as a method but I don't see clearly how it was factored into the results / discussion. It shows up in Table 1 and the methods. Maybe I missed something? Clarify or remove if not used / discussed.

Supplementary material.

Consider, if possible, including a modified version of Figure 1 in the main text. The variation in behavior and diet is quite compelling support for the individual variation. The only suggested change is to move to a generic bear identifier. The actual code used isn't relevant to readers.

The information on individual IDs is fine in supplementary material. Optional issue only for

consideration.

Reviewer #2:

Remarks to the Author:

REVIEW – NATURE

Evaluation:

Pagano et al. conduct an experiment to estimate polar daily energy expenditure during the ice free season in the Hudson Bay. Energy expenditure is linked to the bears' sex/age class, their levels of activity based on direct video observation and 3 axial accelerometer, their body composition estimated by the doubly labelled water method and their movement parameters derived from GPS records. The DEE is also linked to the amount of time spent consuming food mainly terrestrial. Based on the rate of mass loss during the experiment and the amount of body fat, the authors estimate a date at starvation for each bear. The authors highlight the extreme individual variability in polar bear DEE and the challenges in understanding polar bear fasting metabolism. They conclude that this variability of responses to the lack of their primary prey complicates predictions at the population level.

The manuscript is mostly clear and well written and present an impressive amount of experimental work in challenging conditions. My main criticism is linked to the important variability between individual bears and the diversity of behaviour rendering difficult any general conclusion. As the authors point out we are lacking a good understanding of basic physiological mechanisms for fasting polar bears which impairs our comprehension of how they utilise resources (both body reserves and foraging opportunities when preferred prey items are scarce).

I think the manuscript is extremely rich and dense in terms of individual information which should be its focus. Following revision to address points raised below, I support the manuscript for publication.

General thoughts:

This manuscript is based on a very impressive experiment involving 20 wild free-ranging polar bears captured and sampled twice at a 3 weeks interval. This represents an important logistic effort and I commend the authors for targeting bears of 4 different sex and age classes. However, this also entails that the number of bears in each category is small and precludes some statistical modelling and the testing of the influence of the group itself on several response variables. I appreciate the difficulty in increasing the sample size but given the importance of the interindividual variability I would focus on the description of each bear's individual behaviour and its physiological parameters rather than trying to reach conclusion at the groups' level. For example, the group-wise linear regressions in Fig 4a do not seem appropriate given 1) the small number of individuals in subadult males for example and 2) the lack of spread in activity in adult males (and see comments further down).

Along the same lines I am unsure whether the estimation of date of starvation is warranted in this context. I agree with the authors that it seems that the additional caloric intake from terrestrial food sources is unlikely to sustain the bears during a prolonged ice-free period but 1) it seems that the fasting metabolism of polar bears is far more complex than previously thought and hence a linear constant weight decrease might not be appropriate and 2) only one substantial feeding event could be enough to make the difference between survival and starvation.

As a general point, it would be very useful to link each point on the figures to the bears' ID number to facilitate the link to Table 1 that provides a wealth of information.

I provide below some more detailed comments on the text and figures; some comments are also inserted in the main text.

Detailed comments:

The abstract lacks clarity and could be improved (see below). L 21: We measured daily energy expenditure (DEE), diet, behaviour, movement, and changes in body composition in 20 polar bears on land. Please add over which season and the length of the period the bears' parameters were measured over.

L 21-24: unclear sentences. The reference level is hibernation? So 5.2 DEE between hibernation and land use? But the next sentence says "Most bears had DEEs 2 - 4x predicted hibernating rates"

"elevated DDE" compared to hibernation rate?

I would refrain from discussing the risk of starvation in the abstract, I do not think this is a central point in the present article. I think the main point here is to show the variability in terms of behavioural strategy

Discussion:

You could compare the diet observed in the WHB to the one observed on Svalbard (see references in the main text).

L329-348: The explanation as for why bears use more lean body mass than body fat is not straight forward and lacks clarity in this paragraph.

I am not sure that it is possible to compare an injection of pure glucose as in Jansen et al. (2021) to the effect of consuming a variety of terrestrial food (from berries/grass/antlers with high glucose content to water fowl meat with high protein content) ? But if this is the case and the bears consume food and therefore increase the levels of circulating glucose, then they would not be fasting mode longer and would have lower levels of circulating ketones favouring the stored fat mobilisation to supplement locomotion. So this would not explain why the bears lost more lean body mass?

It is likely that the ratio fat / lean body mass in combination to the total amount of fat plays also a role in which part of the body reserve is mobilized and for which category of individual. Pregnant females need a high relative fatness to enter den. Interestingly the 2 non pregnant females had amongst the lowest fatness index and lost a high proportion of their lean body mass (Fig 4).

Polar bears have an hyper specialized diet of lipid and proteins which in humans induces the rise of circulating ketones. In addition these bears are not in hibernating mode (in an hormonal way) and may balance their energy budget between using fat storage they will require to spend the winter or enter den for pregnant females. In this context of energy conservation but not hibernating, it might be beneficial to loose lean body mass (muscle) which are energetically costly to maintain if not used?

The pure body composition might not be the only element at play here and there is likely an hormonal control that differs between hibernation and fasting but vigil mode.

Figures: General: please provide the bears' ID when graphically possible to make it easier to connect to the information provided in table 1. Identify pregnant/non pregnant females.

Fig1. Can you provide each animal's ID so that it is easier to connect with information in table 1.

Fig. 3. Add the number of individuals in each category on the figure itself to facilitate reading (even if it is written in the caption). Identify the data points corresponding to pregnant adult females. How is it possible to construct a boxplot with only 3 points?

Fig 4. a) This figure presenting the predicted lines for each age sex group of the DEE as a function of the activity has a few issues. Although the general tendence seems fairly clear and linear (increase activity = increase DEE), there are some strong differences per age sex group. However, fitting a linear regression for each group seems inappropriate:

- Most groups have a low number of points (n=3, 4, 5) which renders difficult to fit a linear model for each group (or any kind of model).
- The group with most points (adult F) includes 2 non pregnant females and the trend does not seem linear.
- The spread of activity varies between groups, for example, most adult males have an activity between 1 and 6 % and only one point at 20%. The regression line will be mostly influenced by the group of four points.
- It seems strange that the model with interaction between SexAge group and activity is not better supported given the clear differences in slopes between the groups?

b) Add the animals' ID

Fig 6: Same question as above, boxplot with 3 values?

Supplementary figures:

Fig 1: indicate in caption that when the bears are not active they are resting per definition. Explain why there is in some cases large discrepancy in total percentage of activity between accelerometer and video data streams?

Fig 2: Were the small amount of time with direct observation of eating due to the camera duty cycle or because the bears did not engage in eating much?

Fig 3: is really informative regarding the circadian activity rhythm and could be moved to the main text.

Fig 4: How would you explain a relative fatness index of 1 (that is equal mass of fat and lean tissue) for female X33928? I think the most interesting from this figure is that the relative fatness is higher in pregnant females compared to the other groups) because they likely prepare for den entry.

Tables

Tab1. Explain the following column in caption to improve clarity (LBM change). Add the RER<0.7 indicating of ketogenesis or fasting. Add the start date of the experiment for each bear would help to evaluate whether the bears were on land for a similar amount of time at the start of the experiment. Maybe specify sex/age class in a separate column such as it is easier to identify the groups. Also indicate the females that were pregnant vs non pregnant.

Table 1 gives a very impressive amount of information and could be better presented and discussed in the text.

As a comparison, in Derocher et al. 1990, the pre feeding mean U/C= 15.8 when the bears have been fasting for 36d. This means that bears with U/C > 10 can also have been fasting which essentially would represent all the bears in this experiment except for the one with a U/C = 92.1. So the amount and type of items consumed during the experiment was not sufficient or high quality enough to "push" the bears out of fasting. It may be worth noting in the discussion that after a marine mammals-like meal bears in a fasting state return quickly to a low U/C (Derocher 1990).

References

Add:

Seasonal changes in the ratio of serum urea to creatinine in feeding and fasting polar bears 1991. Malcolm A. Ramsay, Ralph A. Nelson, and Ian Stirling. Canadian Journal of Zoology
<https://doi.org/10.1139/z91-048>

Reviewer #3:

Remarks to the Author:

This is a fascinating study, one that builds on earlier work and extends the understanding of polar bear energy budgets into a critical period. The variation in DEE is quite amazing! While I was initially suspected of these results upon reading the ms in detail, I am convinced the data are correct and that, as the authors argue, represent the true variation between individual animals. The methods were solid, so there is no reason to suspect the results. There is a tremendous amount of information in this ms. Many aspects are important, if not critical, for the conservation and management of polar bears. For example, the observation that the bear could not eat a seal at sea! This makes the loss of sea ice even more deadly.

The authors make several quite important points. For example, I couldn't agree more about the importance of understanding the variation among individuals. Most studies focus on population averages, but as the authors point out, selection operates on individuals, not on populations. A relatively recent paper addresses this issue in elephant seals.

Le Boeuf, B., R. Condit, and J. Reiter. 2019. Lifetime reproductive success of northern elephant seals (*Mirounga angustirostris*). Canadian Journal of Zoology 97:1203-1217.

This paper shows tremendous variation in the reproductive success of females and that the population is supported by relatively few successful "super moms".

It was interesting that some of the polar bears used lean tissue during the fast. They suggest that glucose from the consumption of berries down-regulates keto-acid production. The author might also consider the possibility of a thermoregulatory cost, as reported for harp seals.

Worthy, G. A. J., and D. M. Lavigne. 1987. Mass-loss, metabolic rate, and energy utilization by Harp and Gray Seal Pups during the Postweaning Fast. Physiological Zoology 60:352-364.

Specific comments

The authors report rates of energy use kJ/ kg-day; while this is correct, the appropriate SI unit for the rate of energy is Watts/kg. However, I have found that many readers find the use of Watts

confusing, so this is probably more of a Journal issue whether to follow the strict use of SI units.

Fig 4 legend

Lines 162-166. It's hard to follow which line is which. The legend clearly describes the solid black line but does not describe what the other dotted and dash-dot dash lines are. I assume those are to be derived from the legend box inset the figure? It would be clearer if they were described in the figure legend. It's confusing as those data appear to be from other studies. This should be clearly stated, as it leads to confusion.

It's unclear what the predicted date of starvation refers to.

On line 63, it states, "have predicted that up to 24% of the adult males would die of starvation if the summer fast."

However, the next use of the word starvation is on line 220 "Predicted date of starvation."

Is this the predicted date of death by starvation or the initiation of stage three fasting? Stage three fasting refers to when animals have depleted their fat stores and are now burning protein and are in a terminal phase whichever this refers to please clarify.

Reviewer #1 (Remarks to the Author):

An extremely well-designed study that significantly advances our understanding of polar bear ecology and their fasting abilities. One of the most profoundly interesting studies I've read recently. The presentation of the study is very clear. I have some specific comments for consideration but nothing overly significant.

Response: We appreciate the reviewer's feedback, and we are pleased that the reviewer found our study to be informative.

19 – “Declines in Arctic sea ice are increasing polar bear land use.” many people won't understand the use of land by polar bears as a habitat when sea ice is unavailable. Consider clarifying this statement as many people may only read the abstract. Further, “land use” is a bit vague. It may be useful to state that more bears are using land (across their range) but also for longer. Optional change.

Response: We appreciate this reviewer's suggestion. However, given the word limitations for abstracts in Nature Communications (150 words) we have opted to not expand on our meaning of polar bear “land use” as doing so would require removing other details of our study.

35-7 – “and even individual variation” – isn't the amount of energy an individual has stored also individual variation. Some clarification on what you mean on this point would be useful.

Response: Revised to “individual variation in behavior” to clarify what we are referring to here.

Lines 35 – 37: “Such responses likely depend upon the amount of energy reserves an individual has, the duration of the reduced resource period, the energetic tradeoffs between the costs of locating food and the potential energy gained, and even individual variation in behavior.”

39-40 – “inter-individual variation is an often-overlooked driver of animal behavior, energy balance” I'm unconvinced by this statement. Inter-individual variation is a commonly examined metric for many studies (e.g., age, sex, reproductive status, condition, horn size). Some rewording / refinement needed here.

Response: Revised to clarify that we are again referring to inter-individual variation in behavior here as opposed to the other variables the reviewer highlights.

Lines 38 – 40: “The importance of individual variation in species survival and natural selection has been recognized since Darwin⁵, yet inter-individual variation in behavior is an often-overlooked driver of energy balance and population dynamics⁶.”

44 – “increased” vs. “higher” word choice.

Response: Revised to “greater”.

Lines 44 – 45: “Omnivores and other generalist predators often exhibit greater individual variation relative to more specialized predators¹¹.”

45 – I’m a bit uncertain about the “Such variation” – It could be variation within omnivores and generalist predators or the more specialized predators or both collectively. I suspect that latter but because the preceding sentence is set up to show “variation” between the 2 stated groups, consider rewording for clarity. I had to ponder this sentence but couldn’t settle on the intended meaning.

Response: Revised to “This individual variation” to clarify our intended meaning.

Lines 45 – 46: “This individual variation can influence overall population responses to changes in resource availability with implications for ecosystem dynamics⁹.”

51 – “relatively” – relative to what?

Response: We have removed the term “relatively” here.

52 – I’m not convinced that it is weaning of pups that is the trigger for energy reserves – it is the pupping season, weaning, and mating season for seals. It’s more than just the weaning of pups.

Response: Revised to (lines 51 – 52): “when seals are giving birth to and weaning their pups¹⁴.”

68-88 – This section and the predictions is very nicely presented. Concise and well considered.

Response: We are glad the reviewer found this section well presented.

109 – optional but I find the “respectively” unhelpful. I’d rather see 29.5% for adult females, 33.6% for subadult females... It’s easier to keep the 2 parts together.

Response: Revised as suggested.

Lines 109 – 110: “The CV was 29.5 for adult females, 33.6 for subadult females, 21.3 for subadult males, and 59.4% for adult males.”

I may have missed it, but I can't see the study period in the methods nor Table 1. Did the date of sampling affect DEE? I can imagine a bear moving more later in the ice-free period due to migration behavior and not necessarily an individual variation aspect. Specifically, you mention 3-week periods (line 69) suggesting not all sampling of the study animals was concurrent. Please clarify. Somewhere, describe the dates of 1st and last handlings.

Response: Thank you for identifying this oversight. We now provide the date ranges that bears were sampled in each year in the methods,

Lines 403 – 405: “Bears were sampled between 26 August – 14 September 2019, 25 August – 18 September 2021, and 24 August – 21 September 2022.” Sampling dates were consistent across bears and hence unlikely to influence differences in DEE.

116 – because many aspects of polar bear life history are non-linear (i.e., quadratic) did you try a squared term? Mass, litter size, reproductive success and other factors are quadratic which may lead to poor fit.

Response: Per the reviewer's suggestion we now test for potential quadratic effects on DEE and percent body mass change. However, such relationships were not supported.

Lines 532 – 535: “We additionally performed post-hoc tests to evaluate whether any of our continuous variables better described the variation in DEE or percent body mass change as quadratic terms. However, none of these relationships were supported based on AIC_c rankings.”

119-20 – “We found a significant allometric relationship between DEE (MJ•day⁻¹) and mean body mass (kg)” I think mean body mass is probably the mean of the mass at capture and recapture? Please clarify.

Response: We now clarify that this does refer to the mean body mass between the initial capture and recapture.

Lines 119 – 121: “We found a significant allometric relationship between DEE (MJ•day⁻¹) and mean body mass (i.e., the mean body mass between the initial capture and recapture; kg) (DEE = 2.0597 × mass^{0.459}, R² = 0.27, P = 0.05, Fig. 4b).”

Table 1. The second column shows a Bear identifier but I cannot see that this is ever used in the reporting of results. Consider removing this column as it allows focus on the relevant elements. Optional change.

Response: Reviewer 1 and 2 provided conflicting suggestions in this regard. Reviewer 1 suggests removing reference to specific Bear identifiers while Reviewer 2 suggests adding

more references to specific Bear identifiers throughout. Given these conflicting suggestions, we opted to retain reference to the Bear identifiers in Table 1.

Figure 2a & b. The bear is showing up on my copy as partly white & partly in a pale orange. May be a PDF file issue.

Response: The bear shading was intended to reflect the percent activity of each bear. We now clarify this in the figure legend.

Lines 143 – 144: “The bear shading (orange) reflects the overall mean activity expressed as a % total time spent active (non-resting).”

211-13 – Speculation of feeding on a seal or beluga is reasonable but it’s also possible that the bear fed on a caribou or another bear. There are even a few walrus in southern Hudson Bay (or dead bowheads). Cannibalism is an option. I would not be so conclusive – perhaps adding “or other large mammal” or just stating a large mammal was likely fed on. This point is raised again on line 324 “likely fed on a marine mammal on land” – again, speculative. Not a big deal but it is perhaps unnecessary. Especially within the context of “windfall” and the specifics of “marine mammal prey”. Further, on line 327, do you really mean “prey”. I don’t think of scavenged animals necessarily as “prey” within the context of predation. I didn’t see any support for predation vs. scavenging carrion. I find the conclusive nature of the statement unsupported. Same issue on line 369. It is the opportunistic feeding that matters – is it really only marine mammals? A dead moose, caribou, or another polar bear all provide significant energy sources.

Response: While the movements and location of this bear are suggestive of feeding on a marine mammal rather than a terrestrial mammal, we agree that we lack the data to definitively indicate the type of prey this bear consumed. As suggested by the reviewer, we now clarify that this consumption could be from a different species of large mammal.

Lines 214 – 216: “The substantive mass increase, coastal movements, and large change in U/C ratio all suggest he likely fed on a seal, beluga carcass, or other large mammal within a few days of recapture.”

Lines 245 – 247: “The subadult male that gained body mass while onshore initially had a predicted date of starvation that preceded this date of sea ice freeze-up but his predicted date of starvation increased by 62 days as a result of presumably feeding on a large mammal.”

Lines 344 – 347: “In contrast, the one individual that likely fed on a large mammal on land, increased his body mass by 14% (Supplementary Fig. 2), which is similar to the percent mass gain found in female polar bears feeding on ringed seals on the sea ice over 8 – 11 days³⁸.”

Lines 392 – 395: “Nevertheless, declines in body mass were remarkably consistent among 95% of the bears in our study, which emphasizes that none of the energetic strategies were more beneficial for surviving the on-land period, though the foraging response may result in opportunistic feeding events on large mammals.”

Additionally, we have removed the word “prey” as suggested.

Lines 347 – 348: “This highlights the disparity in the energetic windfall polar bears acquire through energy-dense marine mammals relative to terrestrial-based resources^{19,38}.”

299 – wording “ill-equipped” – perhaps “poorly adapted”. Optional.

Response: We believe the term “poorly adapted” may be misinterpreted by some readers. Therefore, we have opted to retain our use of the term “ill-equipped” here.

Lines 320 – 321: “These observations suggest polar bears may be ill-equipped to feed while in water.”

308-11 – for context the mass loss rates reported in this manuscript and cited in Pilfold et al. are very similar to Derocher, A.E., and Stirling, I. 1995. Temporal variation in reproduction and body mass of polar bears in western Hudson Bay. *Can. J. Zool.* 73(9): 1657-1665. doi: 10.1139/z95-197.

The reported mass loss rates in this paper suggest that mass loss rates have not changed much over time. This is not a point made in the paper but possibly worth noting.

Response: We already indicate that our mass loss rates are similar to previous estimates and reference Atkinson et al. 1996.

Lines 327 – 328: “Estimates of daily mass loss were also consistent with previous estimates for WHB polar bears while onshore³⁹.”

We agree that Derocher and Stirling 1995 among others also report similar rates of mass loss of free-ranging polar bears on land in western Hudson Bay. However, due to the journal restrictions on citations (70), we opted to only reference Atkinson et al. 1996 here.

352-4 – Consider reviewing Miller et al. 2022 (*Polar Biology*) who compared starvation threshold of lactating and non-lactating females in the same population. This study supports the “we would expect adult female polar bears with dependent young to have significantly greater DEEs relative to the solitary adult females in our study depending on the extent to which they continue to lactate.” And the vulnerability of nursing females. Optional.

Response: We thank the reviewer for highlighting this reference. We now cite this reference in relation to this sentence.

Lines 379 – 381: “Hence, we would expect adult female polar bears with dependent young to have significantly greater DEEs relative to the solitary adult females in our study depending on the extent to which they continue to lactate.⁵³”

360-60 “Research on survival, reproductive success, and other fitness metrics typically examine population-level responses with limited focus on individual-level variation.” It may be just wording or how one view’s research but I think that many studies on survival and reproductive success focus on individual traits (variation between individuals). This sort of research often focusses on variation in age, horn size, antlers, body mass, or other individual traits. I don’t really agree with the statement. There are hundreds if not thousands of studies on a huge diversity of taxa from guppies to lions.

Response: We now clarify that we are referring to individual variation in behavior here. As opposed to the other variables the reviewer references.

Lines 385 – 386: “Research on survival, reproductive success, and other fitness metrics typically examine population-level responses with limited focus on individual-level variation in behavior.”

Also see our response to the reviewer’s comment regarding lines 39-40 and the comments of Reviewer 3 in this regard.

302 – the estimation of starvation threshold was calculated for polar bears in this population in this paper Miller, E.N., Lunn, N.J., McGeachy, D., and Derocher, A.E. 2022. Autumn migration phenology of polar bears (*Ursus maritimus*) in Hudson Bay, Canada. *Polar Biol.* 45: 1023-1034. doi: 10.1007/s00300-022-03050-3.

This might be a useful reference for comparison.

Response: We now cite this reference as suggested by the reviewer in relation to the potential influence of lactation on starvation thresholds.

Lines 379 – 381: “Hence, we would expect adult female polar bears with dependent young to have significantly greater DEEs relative to the solitary adult females in our study depending on the extent to which they continue to lactate.⁵³”

However, we believe any additional evaluation of the starvation thresholds calculated by Miller et al. 2022 along with Robbins et al. 2012 and Molnár et al. 2010 are beyond the scope of this study.

The methods and analyses are clearly presented and I have few issues.

Response: We are pleased that the reviewer found the methods and analyses to be clearly presented.

409 – The respiratory exchange ratio is outlined as a method but I don't see clearly how it was factored into the results / discussion. It shows up in Table 1 and the methods. Maybe I missed something? Clarify or remove if not used / discussed.

Response: We indicate that the respiratory exchange ratio values are used to convert CO₂ production to metabolic rate.

Lines 461 – 463: “For both equations, we converted CO₂ production to metabolic rate using the mean of the individual-specific RER values from the initial capture and recapture and the equation from Weir⁶⁰.”

Supplementary material.

Consider, if possible, including a modified version of Figure 1 in the main text. The variation in behavior and diet is quite compelling support for the individual variation. The only suggested change is to move to a generic bear identifier. The actual code used isn't relevant to readers.

The information on individual IDs is fine in supplementary material. Optional issue only for consideration.

Response: Per the reviewer's suggestion we have moved the Supplemental Figure 1 to the main text (now Fig. 5). However, Reviewers 1 and 2 provided conflicting suggestions regarding the inclusion of individual IDs. Given Reviewer 2's feedback, we opted to retain reference to the Bear identifiers in this figure within the main text. We will defer to the editor if it would be preferable to exclude these identifiers.

Reviewer #2 (Remarks to the Author):

REVIEW – NATURE

Evaluation:

Pagano et al. conduct an experiment to estimate polar daily energy expenditure during the ice free season in the Hudson Bay. Energy expenditure is linked to the bears' sex/age class, their levels of activity based on direct video observation and 3 axial accelerometer, their body composition estimated by the doubly labelled water method and their movement parameters derived from GPS records. The DEE is also linked to the amount of time spent consuming food mainly terrestrial. Based on the rate of mass loss during the experiment and the amount of body fat, the authors estimate a date at starvation for each bear. The authors highlight the extreme individual variability in polar bear DEE and the challenges in understanding polar bear fasting metabolism. They conclude that this variability of responses to the lack of their primary prey

complicates predictions at the population level.

The manuscript is mostly clear and well written and present an impressive amount of experimental work in challenging conditions. My main criticism is linked to the important variability between individual bears and the diversity of behaviour rendering difficult any general conclusion. As the authors point out we are lacking a good understanding of basic physiological mechanisms for fasting polar bears which impairs our comprehension of how they utilise resources (both body reserves and foraging opportunities when preferred prey items are scarce).

I think the manuscript is extremely rich and dense in terms of individual information which should be its focus. Following revision to address points raised below, I support the manuscript for publication.

Response: We appreciate the reviewer's feedback, and we are pleased that the reviewer found our study to be informative. We agree that the individual information is particularly intriguing and crafted the manuscript to highlight the degree of individual variation we discovered within our dataset. We also believe that the analyses we've presented are important to contextualize the individual variation within our dataset and to assess the overall degree of such variation.

General thoughts:

This manuscript is based on a very impressive experiment involving 20 wild free-ranging polar bears captured and sampled twice at a 3 weeks interval. This represents an important logistic effort and I commend the authors for targeting bears of 4 different sex and age classes. However, this also entails that the number of bears in each category is small and precludes some statistical modelling and the testing of the influence of the group itself on several response variables. I appreciate the difficulty in increasing the sample size but given the importance of the interindividual variability I would focus on the description of each bear's individual behaviour and its physiological parameters rather than trying to reach conclusion at the groups' level. For example, the group-wise linear regressions in Fig 4a do not seem appropriate given 1) the small number of individuals in subadult males for example and 2) the lack of spread in activity in adult males (and see comments further down).

Response: We appreciate the reviewer's concern regarding our limited sample size. The multiple linear regression we performed was done with the intent to evaluate which potential factors influenced DEEs in our study. The model selection approach using AIC_c is designed to balance the number of parameters relative to the overall sample size which was 20 individuals. From that analysis, sex and age class and activity were identified as the best supported variables in influencing DEEs. While we do present a figure of the output from this multiple linear regression to highlight the fit of the data relative to those relationships, we do not provide the associated regression equations as we agree that these equations would be based on relatively small sample sizes. We now further highlight the limited nature of our sample size in the discussion.

Lines 372 – 375: “Although we were able to measure the ecophysiology of four age and sex classes of polar bears while on land, this resulted in small sample sizes for each age and sex class which in combination with the large individual variation we documented, limits our ability to generalize predictions of polar bear energy demands on land among age and sex classes.”

Along the same lines I am unsure whether the estimation of date of starvation is warranted in this context. I agree with the authors that it seems that the additional caloric intake from terrestrial food sources is unlikely to sustain the bears during a prolonged ice-free period but 1) it seems that the fasting metabolism of polar bears is far more complex than previously thought and hence a linear constant weight decrease might not be appropriate and 2) only one substantial feeding event could be enough to make the difference between survival and starvation.

Response: Our analysis of the predicted time to starvation is intended to evaluate how DEE and changes in body composition during the study period influenced the predicted time to starvation using an approach similar to previous research (i.e., Robbins et al. 2012). We agree that such an analysis ignores the complexity of DEE over the entire onshore period as well as the potential for future feeding events to alter the predicted time to starvation. Nevertheless, we believe the analysis provides a method to evaluate whether a fasting or foraging response strategy was more advantageous toward extending the time to starvation. We now further highlight in the methods that this analysis does not account for the potential for bears to modify their DEE while on land or the potential for additional feeding events to extend their date of starvation.

Lines 558 – 560: “This analysis assumes bears maintained their measured DEE while on land and does not account for potential subsequent feeding events post-recapture that could extend their date of starvation.”

Further, as pointed out by Reviewer 1, both the historical data and our data (with the exception of 1 individual) indicate that declines in polar bear body mass are consistent and follow a relatively constant pattern. We address this point in the discussion, lines 327 – 335:

“Estimates of daily mass loss were also consistent with previous estimates for WHB polar bears while onshore³⁹. Pilfold et al.²⁵ reported median mass loss rates of fasting-detained (i.e., kept in a holding facility a median of 17 days) polar bears of 1.4, 1.0, 1.0, and 0.9 kg•day⁻¹ in adult males, solitary adult females, subadult males, and subadult females, respectively, which is similar to the rates of mass loss in our study (1.5, 0.9, 1.0 and 0.7 kg•day⁻¹, respectively). Hence, despite the elevated activity, movement, and food consumption of most bears in our study, mass loss rates were commensurate with rates of mass loss in fasting and relatively inactive bears. This suggests that the more active bears in our study were able to compensate for their elevated DEE through consumption of terrestrial foods.”

As a general point, it would be very useful to link each point on the figures to the bears' ID number to facilitate the link to Table 1 that provides a wealth of information.

Response: Reviewer 1 and 2 provided conflicting suggestions in this regard. Reviewer 1 suggests removing reference to specific Bear identifiers while Reviewer 2 suggests adding more references to specific Bear identifiers throughout. Given these conflicting suggestions, we opted to retain reference to the Bear identifiers in Table 1 and Fig. 5 and added bear identifiers to the figure legend for Fig. 2. However, we believe making individual specific points for Figs. 3, 4, and 6 (now Fig. 8) would be overly distracting for readers and make these figures more difficult to interpret. As an example, we compiled the following revised version of Fig. 3A, which we believe is more challenging to interpret relative to the current version:

I provide below some more detailed comments on the text and figures; some comments are also inserted in the main text.

Detailed comments:

The abstract lacks clarity and could be improved (see below). L 21: We measured daily energy expenditure (DEE), diet, behaviour, movement, and changes in body composition in 20 polar bears on land. Please add over which season and the length of the period the bears' parameters were measured over.

Response: Revised as suggested.

Lines 20 – 22: “We measured daily energy expenditure (DEE), diet, behavior, movement, and changes in body composition in 20 polar bears on land over 19 – 23 days from August to September in Manitoba, Canada.”

L 21-24: unclear sentences. The reference level is hibernation? So 5.2 DEE between hibernation and land use? But the next sentence says “Most bears had DEEs 2 – 4× predicted hibernating rates”

“elevated DDE” compared to hibernation rate?

Response: The 5.2-fold range in DEE referred to the range within our dataset. For clarity, we have removed reference to DEEs relative to predicted hibernating rates.

Lines 22 – 24: “We found a 5.2-fold range in DEE and 19-fold range in activity, from hibernation-like DEEs to levels approaching active bears on sea ice, including three individuals that made energetically demanding swims totaling 54 – 175 km.”

I would refrain from discussing the risk of starvation in the abstract, I do not think this is a central point in the present article. I think the main point here is to show the variability in terms of behavioural strategy

Response: Our reference to predicted time to starvation in the abstract is largely in regard to the loss of body mass bears exhibited in our study. Nevertheless, we believe it is important to address the risk of starvation in the abstract as it highlights the potential significance of our findings on polar bear vital rates while on land, which may not be common knowledge to all readers.

Discussion:

You could compare the diet observed in the WHB to the one observed on Svalbard (see references in the main text).

Response: We believe such an evaluation is beyond the scope of this study particularly given the journal restrictions on citation numbers (70), which we have already exceeded given the references recommended by Reviewers 1 and 3.

L329-348: The explanation as for why bears use more lean body mass than body fat is not straight forward and lacks clarity in this paragraph.

Response: Here we attempted to speculate on some potential physiological mechanisms that could explain the basis for why some of the bears used more lean body mass than body fat. We conclude this paragraph by highlighting that this topic requires further research.

Lines 369 – 372: “Hence, changes in body composition of active polar bears while on land appear to be complex and warrant further research given the implications for overall body condition, energy storage, and predicted time to starvation²¹.”

Based on the comments of this reviewer and Reviewer 3 we have added some additional potential drivers of increased metabolism of lean body mass over fat mass.

Lines 366 – 369: “An alternative mechanism of this increased use of lean body mass over fat mass may be to preserve fat to aid thermoregulation in the fall and winter⁴⁹. Lean body mass is also more energetically costly to maintain than fat mass, which may favor the increased metabolism of lean body mass in active bears in a negative energy balance⁵⁰.”

- 49. Worthy, G. A. J. & Lavigne, D. M. Mass loss, metabolic rate, and energy utilization by harp and gray seal pups during the postweaning fast. *Physiol. Zool.* 60, 352–364 (1987).**
- 50. Wright, T., Davis, R. W., Pearson, H. C., Murray, M. & Sheffield-Moore, M. Skeletal muscle thermogenesis enables aquatic life in the smallest marine mammal. *Science* 373, 223–225 (2021).**

I am not sure that it is possible to compare an injection of pure glucose as in Jansen et al. (2021) to the effect of consuming a variety of terrestrial food (from berries/grass/antlers with high glucose content to water fowl meat with high protein content) ? But if this is the case and the bears consume food and therefore increase the levels of circulating glucose, then they would not be fasting mode longer and would have lower levels of circulating ketones favouring the stored fat mobilisation to supplement locomotion. So this would not explain why the bears lost more lean body mass?

Response: As we indicate in response to the previous comment, here we attempted to speculate on some potential physiological mechanisms that could explain the basis for why some of the bears used more lean body mass than body fat. Our intent here as the reviewer indicates is that feeding bears may have reduced ketone levels similar to Jansen et al. (2021) where they found reduced ketone levels of fed hibernating bears led to reduced free fatty acids and glycerol indicating that lipolysis was reduced (i.e., fat mobilization was reduced), which would suggest that protein would be mobilized to compensate for increased activity.

Lines 359 – 362: “Thus, polar bears feeding on terrestrial foods would likely have suppressed ketone production, resulting in increased appetite and food-seeking behavior. However, the increased energetic cost of foraging when coupled with the increased ketone and glucose uptake by active muscles⁴⁷ would likely reduce lipolysis and exacerbate lean mass loss over fat loss.”

It is likely that the ratio fat / lean body mass in combination to the total amount of fat plays also a role in which part of the body reserve is mobilized and for which category of individual. Pregnant females need a high relative fatness to enter den. Interestingly the 2 non pregnant

females had amongst the lowest fatness index and lost a high proportion of their lean body mass (Fig 4).

Polar bears have an hyper specialized diet of lipid and proteins which in humans induces the rise of circulating ketones. In addition these bears are not in hibernating mode (in an hormonal way) and may balance their energy budget between using fat storage they will require to spend the winter or enter den for pregnant females. In this context of energy conservation but not hibernating, it might be beneficial to loose lean body mass (muscle) which are energetically costly to maintain if not used?

The pure body composition might not be the only element at play here and there is likely an hormonal control that differs between hibernation and fasting but vigil mode.

Response: This is an interesting point, which we have now included in this paragraph,

Lines 367 – 369: “Lean body mass is also more energetically costly to maintain than fat mass, which may favor the increased metabolism of lean body mass over fat mass in active bears in a negative energy balance ⁵⁰.”

50. Wright, T., Davis, R. W., Pearson, H. C., Murray, M. & Sheffield-Moore, M. Skeletal muscle thermogenesis enables aquatic life in the smallest marine mammal. *Science* 373, 223–225 (2021).

Figures: General: please provide the bears’ ID when graphically possible to make it easier to connect to the information provided in table 1. Identify pregnant/non pregnant females.

Response: We believe making individual specific points for Figs. 3, 4, and 6 (now Fig. 8) would be overly distracting for readers (see our previous response in this regard).

However, individual specific data is available in Table 1 as well as in Figs. 5, S1, and S2, and Table S3. As suggested by the reviewer, we now color-code between the 2 non-pregnant females in our dataset and the 6 pregnant females in Figs. 3, 4, and 6 (now Fig. 8).

Fig1. Can you provide each animal’s ID so that it is easier to connect with information in table 1.

Response: We believe providing each animal’s ID would be overly distracting for the broad audience of Nature Communications readers as it would require 20 different labels (see our previous response in this regard).

Fig. 3. Add the number of individuals in each category on the figure itself to facilitate reading (even if it is written in the caption). Identify the data points corresponding to pregnant adult females. How is it possible to construct a boxplot with only 3 points?

Response: We have now added sample sizes to the axis labels for Figs. 3 and 6 (now Fig. 8) as suggested and color-code between the 2 non-pregnant females in our dataset and the 6 pregnant females. Given our limited sample sizes due to the financial and logistic

challenges involved in this study, we opted to visually display our data using boxplots, which indicate the median, 1st and 3rd quartiles, maximum within $1.5\times$ the inter-quartile range, and minimum within $1.5\times$ the inter-quartile range. With a sample size of 3, which occurs in the case of subadult males, the boxplot simply reflects the observational data which is the spread of the three data points.

Fig 4. a) This figure presenting the predicted lines for each age sex group of the DEE as a function of the activity has a few issues. Although the general tendency seems fairly clear and linear (increase activity = increase DEE), there are some strong differences per age sex group. However, fitting a linear regression for each group seems inappropriate:

- Most groups have a low number of points ($n=3, 4, 5$) which renders difficult to fit a linear model for each group (or any kind of model).
- The group with most points (adult F) includes 2 non pregnant females and the trend does not seem linear.
- The spread of activity varies between groups, for example, most adult males have an activity between 1 and 6 % and only one point at 20%. The regression line will be mostly influenced by the group of four points.
- It seems strange that the model with interaction between SexAge group and activity is not better supported given the clear differences in slopes between the groups?

Response: As we indicate in response to the reviewer's main comment in this regard, the multiple linear regression we performed was done with the intent to evaluate which potential factors influenced DEEs in our study. The model selection approach using AIC_c is designed to balance the number of parameters relative to the overall sample size which was 20 individuals. From that analysis, sex and age class and activity were identified as the best supported variables in influencing DEEs. We tested for the effect of pregnancy in our analysis and it was not supported (Table S1). While we do present the output from this multiple linear regression to highlight the fit of the data relative to those relationships, we do not provide the associated regression equations as we agree that these equations are based on relatively small sample sizes. We now further highlight the limited nature of our sample size in the discussion.

Lines 373 – 376: “Although we were able to measure the ecophysiology of four age and sex classes of polar bears while on land, this resulted in small sample sizes for each age and sex class which in combination with the large individual variation we documented, limits our ability to generalize predictions of polar bear energy demands on land among age and sex classes.”

That the interaction between SexAge and activity was not supported was likely related to the limited sample size of our dataset ($n = 20$) and the large number of parameters in such a model ($k = 9$).

b) Add the animals' ID

Response: We believe providing each animal's ID would be overly distracting for the broad audience of Nature Communications readers as it would require 20 different labels (see our previous response in this regard).

Fig 6: Same question as above, boxplot with 3 values?

Response: See our response in regard to Fig. 3.

Supplementary figures:

Fig 1: indicate in caption that when the bears are not active they are resting per definition.

Response: As suggested by the reviewer, we now clarify in the figure legend that activity refers to non-resting behaviors. Per the suggestion of Reviewer 1 this figure has been moved to the main text

Lines 219 – 220: “Fig. 5. Percent time engaged in active (non-resting) behaviors by polar bears on land near Churchill, Manitoba, Canada.”

Explain why there is in some cases large discrepancy in total percentage of activity between accelerometer and video data streams?

Response: As we indicate in the manuscript, the video collars only recorded during daylight hours while the accelerometers recorded continuously.

Lines 204 – 207: “Movement rates, activity, and time spent eating were all greater during daylight hours with time spent eating occurring 2 – 11× more frequently during daylight hours among age and sex classes based on tri-axial accelerometer data (Fig. 6).”

Hence, the activity of bears during daylight hours (as recorded in the video collar data) are greater relative to the overall activity recorded by the accelerometer data.

Fig 2: Were the small amount of time with direct observation of eating due to the camera duty cycle or because the bears did not engage in eating much?

Response: The camera duty cycles were consistent across bears. Hence, as is shown in Supplementary Fig. 2 (now Supplementary Fig. 1) some individuals spent more time eating than other individuals. That being said, the duration of time we were able to quantify as eating reflects a combination of the time each bear spent eating and the amount of video footage that was recorded. We highlight that this figure reflects duty cycled footage in the main text.

Lines 199 – 201: “The amount of time bears were recorded eating within the duty cycled footage ranged from 1 min to 3 hr (Supplementary Fig. 1).”

Fig 3: is really informative regarding the circadian activity rhythm and could be moved to the main text.

Response: Per the reviewer's suggestion, we have moved this figure to the main text (now Fig. 6).

Fig 4: How would you explain a relative fatness index of 1 (that is equal mass of fat and lean tissue) for female X33928? I think the most interesting from this figure is that the relative fatness is higher in pregnant females compared to the other groups) because they likely prepare for den entry.

Response: With a two-compartment model of body composition, an animal's mass consists of fat mass and fat-free mass (lean body). Hence an individual (such as X33934) that is 50% body fat would have a relative fatness index of 1.

Tables

Tab1. Explain the following column in caption to improve clarity (LBM change).

Response: Thank you for identifying this oversight. We now define LBM in the table header.

Lines 129 – 130: “Daily energy expenditure (DEE), respiratory exchange ratio (RER), serum progesterone (P₄), and changes in total body mass, lean body mass (LBM), and fat mass of 20 polar bears on land near Churchill, Manitoba, Canada in 2019, 2021, and 2022.”

Add the RER<0.7 indicating of ketogenesis or fasting.

Response: We now indicate in the Table heading that RERs < 0.7 are believed to reflect ketogenesis while fasting.

Line 133: “RERs < 0.70 are believed to reflect ketogenesis while fasting⁵⁷.”

Add the start date of the experiment for each bear would help to evaluate whether the bears were on land for a similar amount of time at the start of the experiment.

Response: Capture dates of bears were in close proximity to one another and can be extracted from the details we now provide in the methods.

Lines 403 – 405: “Bears were sampled between 26 August – 14 September 2019, 25 August – 18 September 2021, and 24 August – 21 September 2022.”

The specific capture date can further be extracted from the data release associated with this manuscript. Given that this table is already large and contains a lot of data we are

hesitant to add additional columns that would distract from the primary data already included here.

Maybe specify sex/age class in a separate column such as it is easier to identify the groups.

Response: We opted to provide the age of individuals (in years) here rather than listing their age class as we expect some readers will be interested in the spread of ages of the adults that we sampled. We believe adding an additional column to this Table would be distracting given that it is already large and contains a lot of data.

Also indicate the females that were pregnant vs non pregnant.

Response: We provide the serum progesterone (P_4) levels in this table and indicate that serum P_4 levels $> 2.5 \text{ ng}\cdot\text{ml}^{-1}$ are considered indicative of pregnancy in polar bears in autumn (Derocher et al. 1992). Hence, the pregnancy status of these individuals can already be discerned in this table.

Line 132: “Serum P_4 levels $> 2.5 \text{ ng}\cdot\text{ml}^{-1}$ are considered indicative of pregnancy in polar bears in autumn³⁶.”

Table 1 gives a very impressive amount of information and could be better presented and discussed in the text.

Response: We are pleased that the reviewer found this table to be informative. We reference this table in the manuscript on 7 separate occasions and much of this data is further presented in the subsequent figures. While there are additional details that could be explored through this table, we attempted to highlight only the key findings from this table given the broad readership of the journal.

As a comparison, in Derocher et al. 1990, the pre feeding mean $U/C = 15.8$ when the bears have been fasting for 36d. This means that bears with $U/C > 10$ can also have been fasting which essentially would represent all the bears in this experiment except for the one with a $U/C = 92.1$. So the amount and type of items consumed during the experiment was not sufficient or high quality enough to “push” the bears out of fasting. It may be worth noting in the discussion that after a marine mammals-like meal bears in a fasting state return quickly to a low U/C (Derocher 1990).

Response: Per the reviewer’s suggestion, we have revised the cutoff of fasting to be ≤ 16 .

Lines 132 – 133: “Blood serum urea/creatinine ratios (U/C) ≤ 16 are considered to be an indicator of fasting for > 1 week³⁷.”

There is much subjectivity in the literature regarding the appropriate cutoff point at which U/C ratios are indicative of fasting in bears, which is beyond the scope of this study. For example, Cherry et al. 2009 considered polar bears to be fasting once their U/C ratios were ≤ 10 . For our purposes, the appropriate cutoff point at which U/C ratios are indicative of fasting (i.e., ≤ 10 vs ≤ 16) has no influence on our conclusions.

References

Add:

Seasonal changes in the ratio of serum urea to creatinine in feeding and fasting polar bears 1991. Malcolm A. Ramsay, Ralph A. Nelson, and Ian Stirling. Canadian Journal of Zoology <https://doi.org/10.1139/z91-048>

Response: We have opted not to add this additional reference as we already reference Derocher et al. 1990 which provides a detailed background on the use of urea-creatinine ratios as an indicator of fasting in polar bears. Additionally, we have already exceeded the journal's limit on citation numbers (70) given the references recommended by Reviewers 1 and 3.

Reviewer #3 (Remarks to the Author):

This is a fascinating study, one that builds on earlier work and extends the understanding of polar bear energy budgets into a critical period. The variation in DEE is quite amazing! While I was initially suspected of these results upon reading the ms in detail, I am convinced the data are correct and that, as the authors argue, represent the true variation between individual animals. The methods were solid, so there is no reason to suspect the results. There is a tremendous amount of information in this ms. Many aspects are important, if not critical, for the conservation and management of polar bears. For example, the observation that the bear could not eat a seal at sea! This makes the loss of sea ice even more deadly.

Response: We appreciate the reviewer's feedback, and we are pleased that they found our study to be insightful.

The authors make several quite important points. For example, I couldn't agree more about the importance of understanding the variation among individuals. Most studies focus on population averages, but as the authors point out, selection operates on individuals, not on populations. A relatively recent paper addresses this issue in elephant seals.

Le Boeuf, B., R. Condit, and J. Reiter. 2019. Lifetime reproductive success of northern elephant seals (*Mirounga angustirostris*). Canadian Journal of Zoology 97:1203-1217.

This paper shows tremendous variation in the reproductive success of females and that the population is supported by relatively few successful "super moms".

It was interesting that some of the polar bears used lean tissue during the fast. They suggest that glucose from the consumption of berries down-regulates keto-acid production. The author might also consider the possibility of a thermoregulatory cost, as reported for harp seals.

Worthy, G. A. J., and D. M. Lavigne. 1987. Mass-loss, metabolic rate, and energy utilization by

Harp and Gray Seal Pups during the Postweaning Fast. *Physiological Zoology* 60:352-364.

Response: This is an interesting suggestion. Although we are unable to evaluate whether thermoregulation plays a role in the rate at which polar bears metabolize fat versus lean body mass while on land, we now highlight this as a potential factor.

Lines 366 – 367: “An alternative mechanism of this increased use of lean body mass over fat mass may be to preserve fat to aid thermoregulation in the fall and winter ⁴⁹.”

49. Worthy, G. A. J. & Lavigne, D. M. Mass loss, metabolic rate, and energy utilization by harp and gray seal pups during the postweaning fast. *Physiol. Zool.* 60, 352–364 (1987).

Specific comments

The authors report rates of energy use kJ/ kg-day; while this is correct, the appropriate SI unit for the rate of energy is Watts/kg. However, I have found that many readers find the use of Watts confusing, so this is probably more of a Journal issue whether to follow the strict use of SI units.

Response: For consistency with most other metabolic studies (including previous metabolic studies on polar bears) (e.g., Costa et al. 1986, Speakman et al. 2003, Nagy 2005, Pagano et al. 2018, Molnar et al. 2020), we report metabolic rates in Joules.

Costa, D., and B. Boeuf. 1986. The energetics of lactation in the northern elephant seal, *Mirounga angustirostris*. *Journal of Zoology* 209:21–33.

Molnár, P. K., C. M. Bitz, M. M. Holland, J. E. Kay, S. R. Penk, and S. C. Amstrup. 2020. Fasting season length sets temporal limits for global polar bear persistence. *Nature Climate Change* 10:732–738.

Nagy, K. A. 2005. Field metabolic rate and body size. *Journal of Experimental Biology* 208:1621–1625.

Pagano, A. M., G. M. Durner, K. D. Rode, T. C. Atwood, S. N. Atkinson, E. Peacock, D. P. Costa, M. A. Owen, and T. M. Williams. 2018. High-energy, high-fat lifestyle challenges an Arctic apex predator, the polar bear. *Science* 359:568–572.

Speakman, J. R., T. Ergon, R. Cavanagh, K. Reid, D. M. Scantlebury, and X. Lambin. 2003. Resting and daily energy expenditures of free-living field voles are positively correlated but reflect extrinsic rather than intrinsic effects. 100:14057–14062.

Fig 4 legend

Lines 162-166. It's hard to follow which line is which. The legend clearly describes the solid black line but does not describe what the other dotted and dash-dot dash lines are. I assume those are to be derived from the legend box inset the figure? It would be clearer if they were described

in the figure legend. It's confusing as those data appear to be from other studies. This should be clearly stated, as it leads to confusion.

Response: We have corrected this figure legend and clarified which line refers to which reference in the legend itself in addition to the legend box.

Lines 165 – 170: “(b) the allometric regression (solid line) of DEE with mean body mass compared to the DEE of female polar bears on the spring sea ice in the Beaufort Sea (white points and dashed line)³⁸, predicted DEE of male polar bears on land in Western Hudson Bay based on changes in body composition (white squares)³⁹, predicted basal metabolic rates (BMR; dotted line)³³, and the average energetic cost of hibernation in Holarctic bears (dash-dot line)²⁴.”

It's unclear what the predicted date of starvation refers to.

On line 63, it states, “have predicted that up to 24% of the adult males would die of starvation if the summer fast.”

However, the next use of the word starvation is on line 220 “Predicted date of starvation.”

Is this the predicted date of death by starvation or the initiation of stage three fasting? Stage three fasting refers to when animals have depleted their fat stores and are now burning protein and are in a terminal phase whichever this refers to please clarify.

Response: All of the predicted dates of starvation refer to the predicted date of death by starvation. We now clarify this in the text.

Lines 240 – 242: “We found no significant difference in the predicted date of death by starvation between the initial capture and recapture (mean difference = -5.0 days, $t = -1.2$ days, $P = 0.2$, Fig. 8) based on each bear's body composition and DEE.”

Lines 541 – 542: “Lastly, we predicted the time to death by starvation for each bear based on its body composition at initial capture and recapture and its average DEE.”

Review by Gregory Thiemann, York University

15 September 2023

I am grateful for the opportunity to review this manuscript. The study broadly investigates how polar bears Western Hudson Bay cope with food scarcity associated with the annual ice-free season. It examines individual behavioural and physiological patterns in 20 bears during a season where, it has long been assumed, bears adopt an energy-conserving strategy of limited energetic intake and output. Although some bears in the study met this assumption, there was greater diversity in individual responses than has previously been recognized.

The study represents a major advancement in our understanding of the ecology of polar bears in Western Hudson Bay. The study was thoughtfully conceived and executed. It provides critical new insights into the degree of individual variation in polar bear responses to food restriction. It also establishes quantitative values for processes (e.g., activity, movement, feeding, compositional mass loss) that were previously understood mostly qualitatively. The work will have important implications not only for predicting population-level responses to climate warming and sea ice decline, but also understanding individual variation in all aspects of polar bear ecology.

I have provided some minor editorial suggestions in the attached Word file, but have no substantive recommendations for improving the manuscript. I would recommend it for publication in its current form and congratulate the authors on an excellent piece of work.

Response: We appreciate the reviewer's feedback and we're pleased that the reviewer found our study to be informative.

Line 19: Change "Declines in" to "Declining".

Response: Revised as suggested.

Lines 19 – 20: "Declining Arctic sea ice is increasing polar bear land use."

Lines 21 – 22: I think the study location is essential to any Abstract, so offered some suggestions on how you could possibly squeeze it in.

Response: Revised as suggested.

Lines 20 – 22: “We measured daily energy expenditure (DEE), diet, behavior, movement, and changes in body composition in 20 polar bears on land over 19 – 23 days from August to September in Manitoba, Canada.”

Line 91, Fig. 1: Explain the shading of the circles.

Response: We now explain in the figure legend that light circles represent capture locations while dark circles represent recapture locations.

Lines 91 – 94: “Capture (light circles) and recapture (dark circles) locations and GPS movement paths of 20 polar bears (8 adult females (green lines), 4 subadult females (orange lines), 3 subadult males (yellow lines), and 5 adult males (purple lines)) dosed with doubly-labeled water and equipped with GPS-enabled video camera collars on land near Churchill, Manitoba, Canada.”

Line 99: I think it’s worth specifying that no females had cubs, add “solitary” here.

Response: Revised as suggested.

Lines 98 – 100: “We measured the energy expenditure, diet, behavior, activity, movement rate, blood chemistry, and body composition of 20 polar bears: 8 solitary adult females, 5 adult males, 4 subadult females, and 3 subadult males (Fig. 1, Table 1)³².”

Line 143, Fig. 2: Add “the proportion of time spent active”.

Response: We have now added a similar sentence in response to Reviewer 1’s comment on this figure.

Lines 143 – 144: “The bear shading (orange) reflects the overall mean activity expressed as a % total time spent active (non-resting).”

Line 152, Fig. 3: Add “expressed as % total time spent active(?)”.

Response: Revised as suggested.

Lines 153 – 154: “(c) accelerometer-derived activity expressed as a % total time spent active (non-resting)”.

Line 178: Remove the word “relatively”.

Response: Revised as suggested.

Line 179: I’d suggest rephrasing to avoid the double negative.

Response: Revised as suggested.

Lines 182 – 184: “Despite large variation in activity and DEE, percent changes in body mass were similar across sex and age classes, and 19 of the 20 bears lost 4 – 11% ($\bar{x} = 7.4 \pm 0.5\%$) of their body mass.”

Line 235 – 236: Might want to make this line a bit heavier, or change the pattern, to make it stand out from the other dashed lines.

Response: Revised by making this line heavier as suggested.

Reviewers' Comments:

Reviewer #1:

Remarks to the Author:

I am fully satisfied by the revisions made to the manuscript. I look forward to seeing the final version in print.

Andrew Derocher

Reviewer #2:

None

Reviewer #3:

Remarks to the Author:

I am satisfied with the revisions the authors have made to this manuscript. I look forward to seeing it published.

Best

Dan Costa